# Average Certified Radius is a Poor Metric for Randomized Smoothing

Chenhao Sun [* 1]  Yuhao Mao [* 1]  Mark Niklas Müller [1 2]  Martin Vechev [1]

## Abstract

Randomized smoothing (RS) is popular for providing certified robustness guarantees against adversarial attacks. The average certified radius (ACR) has emerged as a widely used metric for tracking progress in RS. However, in this work, for the first time we show that ACR is a poor metric for evaluating robustness guarantees provided by RS. We theoretically prove not only that a trivial classifier can have arbitrarily large ACR, but also that ACR is extremely sensitive to improvements on easy samples. In addition, the comparison using ACR has a strong dependence on the certification budget. Empirically, we confirm that existing training strategies, though improving ACR, reduce the model's robustness on hard samples consistently. To strengthen our findings, we propose strategies, including explicitly discarding hard samples, reweighing the dataset with approximate certified radius, and extreme optimization for easy samples, to replicate the progress in RS training and even achieve the state-of-the-art ACR on CIFAR-10, without training for robustness on the full data distribution. Overall, our results suggest that ACR has introduced a strong undesired bias to the field, and its application should be discontinued in RS. Finally, we suggest using the empirical distribution of $p_A$, the accuracy of the base model on noisy data, as an alternative metric for RS.

## 1. Introduction

Adversarial robustness, namely the ability of a model to resist arbitrary small perturbations to its input, is a critical property for deploying machine learning models in security-sensitive applications. Due to the incompleteness

*Equal contribution [1]Department of Computer Science, ETH Zurich, Switzerland [2]LogicStar.ai. Correspondence to: <chensun@student.ethz.ch, {yuhao.mao, martin.vechev}@inf.ethz.ch, mark@logicstar.ai>.

*Proceedings of the 42$^{nd}$ International Conference on Machine Learning*, Vancouver, Canada. PMLR 267, 2025. Copyright 2025 by the author(s).

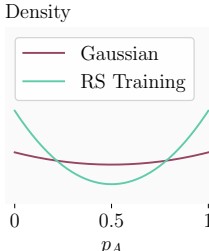

Figure 1: Conceptual illustration of the effect of RS training.

of adversarial attacks which try to construct a perturbation that manipulates the model (Athalye et al., 2018), certified defenses have been proposed to provide robustness guarantees. While deterministic certified defenses (Gowal et al., 2018; Mirman et al., 2018; Shi et al., 2021; Müller et al., 2023; Mao et al., 2023; 2024a; De Palma et al.; Balauca et al., 2024) incur no additional cost at inference-time, randomized certified defenses scale better with probabilistic guarantees at the cost of multiplied inference-time complexity. The most popular randomized certified defense is Randomized Smoothing (RS) (Lécuyer et al., 2019; Cohen et al., 2019), which certifies a smoothed classifier given the accuracy of a base model on noisy data, $p_A$.

To train better models with larger certified radius under RS, many training strategies have been proposed (Cohen et al., 2019; Salman et al., 2019; Jeong & Shin, 2020; Zhai et al., 2020; Jeong et al., 2023). Average Certified Radius (ACR), defined to be the average of the certified radii over each sample in the dataset, has been a common metric to evaluate the effectiveness of these methods. Although some studies refrained from using ACR in their evaluation unintentionally (Carlini et al., 2023; Xiao et al., 2022), the inherent limitation of ACR remains unknown. As a result, the community remains largely unaware of these issues, resulting in subsequent research relying on ACR continuously (Zhang et al., 2023; Jeong & Shin, 2024). However, in this work, we show that ACR is a poor metric for evaluating the true robustness of a given model under RS.

**Main Contributions**

- We theoretically prove that with a large enough certification budget, ACR of a trivial classifier can be arbitrarily large, and that with the certification budget com-

monly used in practice, an improvement on easy inputs contributes much more to ACR than on hard inputs, more than 1000x in the extreme case (§4.1 and §4.2).

- We empirically compare RS training strategies to Gaussian training and show that all current RS training strategies are actually reducing the accuracy on hard inputs where $p_A$ is relatively small, and only focus on easy inputs where $p_A$ is very close to 1 to increase ACR (§4.3). Figure 1 conceptually visualizes this effect.

- Based on these novel insights, we develop strategies to replicate the advances in ACR by encouraging the model to focus only on easy inputs. Specifically, we discard hard inputs during training, reweight the dataset with their contribution to ACR, and push $p_A$ extremely close to 1 for easy inputs via adversarial noise selection. With these simple modifications to Gaussian training, we achieve the new SOTA in ACR on CIFAR-10 and a competitive ACR on IMAGENET without targeting robustness for the general case (§5 and §6).

Overall, our work proves that the application of ACR should be discontinued in the evaluation of RS. In particular, we suggest using the empirical distribution of $p_A$ as more informative metrics, and encourage the community to evaluate RS training more uniformly (§7). We hope this work will motivate the community to permanently abandon ACR and inspire future research in RS.

## 2. Related Work

Randomized Smoothing (RS) is a defense against adversarial attacks that provides certified robustness guarantees (Lécuyer et al., 2019; Cohen et al., 2019). However, to achieve strong certified robustness, special training strategies tailored to RS are essential. Gaussian training, which adds Gaussian noise to the original input, is the most common strategy, as it naturally aligns with RS (Cohen et al., 2019). Salman et al. (2019) propose to add adversarial attacks to Gaussian training, and Li et al. (2019) propose a regularization to control the stability of the output. Salman et al. (2020) further shows that it is possible to exploit a pretrained non-robust classifier to achieve strong RS certified robustness with input denoising. Afterwards, Average Certified Radius (ACR), the average of RS certified radius over the dataset, is commonly used to evaluate RS training: Zhai et al. (2020) propose an attack-free mechanism that directly maximizes certified radii; Jeong & Shin (2020) propose a regularization to improve the prediction consistency; Jeong et al. (2021) propose to calibrate the confidence of smoothed classifier; Horváth et al. (2022) propose to use ensembles as the base classifier to reduce output variance; Vaishnavi et al. (2022) apply knowledge transfer on the base classifier; Jeong et al. (2023) distinguishes hard and

easy inputs and apply different loss for each class. While these methods all improve ACR, this work shows that ACR is a poor metric for robustness, and that these training algorithms all introduce undesired side effects. This work is the first to question the development of RS training strategies evaluated with ACR and suggests new metrics as alternatives. While there exists various ways for RS certification (Cullen et al., 2022; Li et al., 2022), this work only focus on the certification algorithm proposed by Cohen et al. (2019), which is the most widely used in the literature.

To better understand the application of ACR, we conduct a literature review in RS for publications on top conferences (ICML, ICLR, AAAI, NeurIPS), ranging from 2020 (since the first evaluation with ACR) to 2024 (until this work). We only include works that use the certification algorithm proposed by Cohen et al. (2019) and propose general-purpose dataset-agnostic RS algorithms. We identify 13 works with the specified criteria, and report for each of them (1) whether ACR is evaluated, (2) whether universal improvement in certified accuracy is achieved at all radii, (3) whether a customized base model is used (either parameter tuning or architecture design) and (4) whether SOTA achievement is claimed. The result is shown in Table 1. Among them, we find 10 works that customize the model and claim SOTA, which is the focus of our study. 8 of them evaluate with ACR, and 7 of them claim SOTA solely based on ACR, i.e., claim SOTA without universal improvement in certified accuracy at various radii. Therefore, we conclude that ACR is of great importance to the field, and the practice of claiming SOTA based on ACR is widely spread.

## 3. Background

In this section, we briefly introduce the background required for this work.

**Adversarial Robustness** is the ability of a model to resist arbitrary small perturbations to its input. Formally, given an input set $S(x)$ and a model $f$, $f$ is adversarially robust within $S(x)$ iff for every $x_1, x_2 \in S(x)$, $f(x_1) = f(x_2)$. In this work, we focus on the $L_2$ neighborhood of an input, i.e., $S(x) := B_\epsilon(x) = \{x' \mid \|x - x'\|_2 \leq \epsilon\}$ for a given $\epsilon \geq 0$. For a given $(x, y)$ from the dataset $(\mathcal{X}, \mathcal{Y})$, $f$ is robustly correct iff $\forall x' \in S(x), f(x') = y$.

**Randomized Smoothing** constructs a smooth classifier $\hat{f}(x)$ given a base classifier $f$, defined as follows: $\hat{f}(x) := \arg\max_{c \in \mathcal{Y}} \mathbb{P}_{\delta \sim \mathcal{N}(0, \sigma^2 I)}(f(x + \delta) = c)$. Intuitively, the smooth classifier assigns the label with maximum probability in the neighborhood of the input. With this formulation, Cohen et al. (2019) proves that $\hat{f}$ is adversarially robust within $B_{R(x, p_A)}(x)$ when $p_A \geq 0.5$, where $R(x, p_A) := \sigma \Phi^{-1}(p_A)$, $\Phi$ is the cumulative distribution

| Literature | Use ACR | Universal improvement | Claim SOTA | Customized model |
|---|---|---|---|---|
| Zhai et al. (2020) | ✔ | | ✔ | ✔ |
| Jeong & Shin (2020) | ✔ | | ✔ | ✔ |
| Awasthi et al. (2020) | | | | ✔ |
| Jeong et al. (2021) | ✔ | | ✔ | ✔ |
| Horváth et al. (2022) | ✔ | | ✔ | ✔ |
| Chen et al. (2022) | ✔ | | ✔ | ✔ |
| Vaishnavi et al. (2022) | ✔ | | | ✔ |
| Yang et al. (2022) | ✔ | ✔ | ✔ | ✔ |
| Carlini et al. (2023) | | | | ✔ |
| Jeong et al. (2023) | ✔ | | ✔ | ✔ |
| Wu et al. (2023) | | ✔ | ✔ | ✔ |
| Li et al. (2024) | | ✔ | ✔ | ✔ |
| Wang et al. (2024) | ✔ | | ✔ | ✔ |

Table 1: A literature survey in Randomized Smoothing for publications on top conferences, ranging from 2020 to 2024. It only includes works that use the certification method proposed by Cohen et al. (2019) and propose general-purpose dataset-agnostic RS algorithms.

function of the standard Gaussian distribution $\mathcal{N}(0,1)$, and $p_A := \max_{c \in \mathcal{Y}} \mathbb{P}_{\delta \sim \mathcal{N}(0,\sigma^2 I)}(f(x+\delta) = c)$ is the probability of the most likely class. Average Certified Radius (ACR) is defined as the average of $R(x, p_A)\mathbf{1}(\hat{f}(x) = y)$ over the dataset, where $\mathbf{1}(\cdot)$ is the indicator function. In practice, $p_A$ cannot be computed exactly, and an estimation $\hat{p}_A$ such that $\mathbb{P}(p_A \geq \hat{p}_A) \geq 1 - \alpha$ is substituted, where $\alpha$ is the confidence threshold. Thus, $\hat{p}_A$ is computed based on $N$ trials for the event $\mathbf{1}(f(x+\delta) = c)$. We call $N$ the certification budget, which is the number of queries to the base classifier $f$ to estimate $p_A$. Since RS certifies robustness based on the accuracy of the base model on samples perturbed by Gaussian noise, *Gaussian training*, which augments the train data with Gaussian noise, is the most intuitive method to train the base model for RS. Specifically, it optimizes

$$\arg\min_{\theta} \mathbb{E}_{(x,y)\sim(\mathcal{X},\mathcal{Y})} \frac{1}{m} \sum_{i=1}^{m} L(x + \delta_i; y)$$

where $\delta_i$ are sampled from $\mathcal{N}(0, \sigma^2 I)$ uniformly at random, $m$ is the number of samples and $L$ is the cross entropy loss. Due to space constraints, we omit the details of other training strategies for RS, and refer interested readers to Cohen et al. (2019); Salman et al. (2019); Jeong & Shin (2020); Zhai et al. (2020); Jeong et al. (2023).

## 4. Weakness of ACR and the Consequence

We now first theoretically show that ACR can be arbitrarily large for a trivial classifier, assuming enough budget for certification (§4.1). We then demonstrate that with a realistic certification budget, an improvement on easy samples could contribute more than 1000 times to ACR than the same magnitude of improvement on hard samples (§4.2).

Finally, we empirically show that due to the above weakness of ACR, all current RS training strategies evaluated with ACR reduce the accuracy on hard samples and only focus on easy samples (§4.3), improving ACR at the cost of performance on hard samples.

### 4.1. Trivial Classifier with Infinite ACR

In Theorem 1 below we show that, for every classification problem, there exists a trivial classifier with infinite ACR given enough budget for certification, while such a classifier can only robustly classify samples from the most likely class in the dataset and always misclassifies samples from other classes.

**Theorem 1.** For every $M > 0$ and $\alpha > 0$, there exists a trivial classifier $f$ which always predicts the same class with a certification budget $N > 0$, such that the ACR of $f$ is greater than $M$ with confidence at least $1 - \alpha$.

The proof of Theorem 1 is deferred to App. A.1. Despite an intuitive proof, Theorem 1 reveals an important fact: ACR can be arbitrarily large for a classifier with large enough certification budget. Further, a trivial classifier can achieve infinite ACR with minimal robustness on all but one classes. Therefore, the ACR metric presented in the literature is heavily associated with the certification budget; when the certification budget changes, the order of ACR between models may change as well, since inputs with lower $p_A$ require a larger budget to certify. For example, given a perfectly balanced binary dataset and $N = 50$, $\alpha = 0.001$, $\sigma = 1$, a trivial classifier can achieve ACR equals 0.565, while another classifier with $p_A = 0.9$ on every input can only achieve ACR equals 0.544, making the trivial classifier the better one. However, when $N$ is increased to 100, the latter achieves ACR equals 0.756, sur-

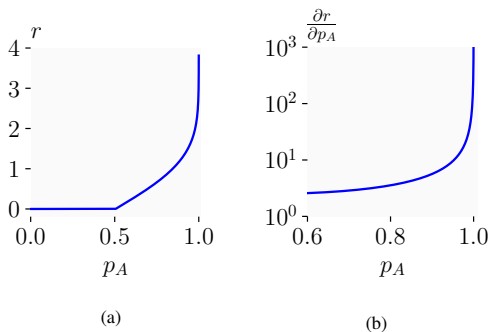

(a)                              (b)

Figure 2: Certified radius $r$ and its sensitivity $\frac{\partial r}{\partial p_A}$ against $p_A$. Note the log scale of y axis in Figure 2b. $N$ is set to $10^5$, $\alpha$ is set to 0.001, and $\sigma$ is set to 1.

| Method | ACR | easy | hard | easy / hard |
|---|---|---|---|---|
| Gaussian | 0.56 | 10.10 | 22.67 | 0.45 |
| SmoothAdv | 0.68 | 5.60 | 5.62 | 1.00 |
| Consistency | 0.72 | 14.99 | 19.32 | 0.78 |
| SmoothMix | 0.74 | 11.72 | 11.79 | 0.99 |
| CAT-RS | 0.76 | 30.45 | 7.12 | 4.28 |

Table 2: The average model parameters gradient $l_2$ norm of easy ($p_A > 0.5$) and hard ($p_A < 0.5$) samples for models trained with different algorithms and $\sigma = 0.5$, along with their relative magnitude (*easy / hard*). The corresponding ACR is provided for reference.

passing the trivial classifier which has ACR equals 0.750. Interestingly, when we further increase $N$ to 200, the trivial classifier achieves ACR equals 0.913, surpassing the latter which has ACR equals 0.909, thus the order is reversed again. This toy example showcases the strong dependence of the evaluation results on the certification budget, especially the comparative order of different models. This is problematic in practice, as the budget usually differs depending on the task and the model, resulting in uncertainty when choosing training strategies evaluated based on ACR.

### 4.2. ACR Prefers Improving Easy Samples

We now discuss the effect of ACR with a constant budget and fixed $\alpha$. We follow the standard certification setting in the literature, setting $N = 10^5$ and $\alpha = 0.001$ (Cohen et al., 2019). With this budget, the maximum certified radius for a data point is $R(x, p_A = 1) = \sigma \Phi^{-1}(\alpha^{1/N}) \approx 3.8\sigma$. However, we will show that $\frac{\partial R(x,p_A)}{\partial p_A}$ grows extremely fast, exceeding $1000\sigma$ when $p_A \to 1$ and close to 0 when $p_A \to 0.5$.

Without loss of generality, we set $\sigma = 1$ and denote $R(x, p_A)$ as $r$. Figure 2 shows $r$ and $\frac{\partial r}{\partial p_A}$ against $p_A$. While $r$ remains bounded by a constant 3.8, $\frac{\partial r}{\partial p_A}$ grows extremely fast when $p_A$ approaches 1. As a result, increasing $p_A$ from 0.99 to 0.999 improves $r$ from 2.3 to 3.0, matching the improvement achieved by increasing $p_A$ from 0 to 0.76. Further, when $p_A < 0.5$, the data point will not contribute to ACR at all, thus optimizing ACR will not increase $p_A$ with a local optimization algorithm like gradient descent. Therefore, it is natural for RS training to disregard inputs with $p_A < 0.5$, as their ultimate goal is to improve ACR.

We use the same toy example as in §4.1 to further demonstrate this problem. If we further increase $N$ beyond 200, we can find, surprisingly, that the classifier with $p_A = 0.9$ for every input always has a smaller ACR than the trivial classifier. This means that achieving $p_A = 1$ on only half of the dataset and getting $p_A = 0$ for the rest is "more robust"

than achieving $p_A = 0.9$ on every input when evaluated with ACR, which is misaligned with our intuition. This toy example highlights again that the unmatched contribution to ACR from easy and hard samples leads to biases.

It is important to note that, while improvements in $p_A$ for easy samples contribute disproportionately more to the overall ACR compared to improvements on hard samples, this does not imply that increasing $p_A$ for easy samples itself is easier. However, we hypothesize that focus on improving $p_A$ on easy samples can increase ACR. As we will show in §5 and §6, this hypothesis holds true empirically.

### 4.3. Selection Bias in Existing RS Training Algorithms

We have shown that ACR strongly prefers easy samples in §4.2. However, since ACR is not differentiable with respect to the model parameters because it is based on counting, RS training strategies usually do not directly apply ACR as the training loss. Instead, they optimize various surrogate objectives, and finally evaluate the model with ACR. Thus, it is unclear whether and to what extent the design of training algorithms is affected by the ACR metric. We now empirically quantify the effect, confirming that the theoretical weakness of ACR has introduced strong selection biases in existing RS training algorithms. Specifically, we show that SOTA training strategies reduce $p_A$ of hard samples and put more weight (measured by gradient magnitude) on easy samples when compared to Gaussian training.

Figure 3 shows the empirical cumulative distribution of $p_A$ for models trained with SOTA algorithms and Gaussian training on CIFAR-10. Clearly, for various $\sigma$, SOTA algorithms have higher density than Gaussian training at $p_A$ close to zero and one. While they gain more ACR due to the improvement on easy samples, hard samples are consistently underrepresented in the final model compared to Gaussian training. The same results hold on IMAGENET (Figure 11 in App. E.2). As a result, Gaussian training has higher $\mathbb{P}(p_A \geq 0.5)$ (clean accuracy), and SOTA algorithms exceed Gaussian training only when $p_A$ passes a certain threshold, i.e., when the certified radius is relatively

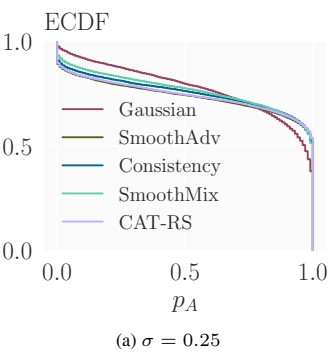

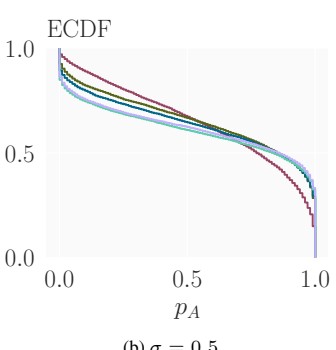

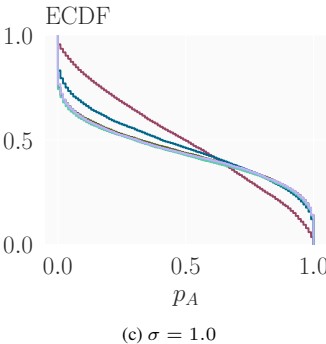

(a) $\sigma = 0.25$        (b) $\sigma = 0.5$        (c) $\sigma = 1.0$

Figure 3: The empirical cumulative distribution (ECDF) of $p_A$ on CIFAR-10 for models trained and certified with various $\sigma$ with different training algorithms.

large. This is problematic in practice, indicating that ACR does not properly measure the model's ability even with a fixed certification setting. For example, a face recognition model could have a high ACR but consistently refuse to learn some difficult faces, which is not acceptable in real-world applications.

To further quantify the relative weight between easy and hard samples indicated by each training algorithm, we measure the average gradient $l_2$ norm of easy and hard samples for model parameters trained with different algorithms and $\sigma = 0.5$, as a proxy for the sample weight. Intuitively, samples with larger gradients on parameters contribute more to training and thus are more important to the final model. As shown in Table 2, Gaussian training puts less weight on easy samples than hard samples, which is natural as easy samples have smaller loss values. However, SOTA algorithms put more weight on easy samples compared to Gaussian training, e.g., the relative weight between easy and hard samples is 4.28 for the best ACR baseline, CAT-RS (Jeong et al., 2023), while for Gaussian training it is 0.45. This confirms that SOTA algorithms indeed prioritize easy samples over hard samples, consistent with the theoretical analysis in §4.2.

## 5. Replicating the Progress in ACR

In §4, we find that ACR strongly prefers easy samples, and this bias has been introduced to the design of RS training strategies. This questions that whether explicitly focusing on easy data during training can effectively replicate the increase in ACR gained by years spent on the development of RS training strategies. In this section, we propose three intuitive modifications on the Gaussian training, answering this question affirmatively.

### 5.1. Discard Hard Data During Training

Samples with low $p_A$ contribute little to ACR, especially those with $p_A < 0.5$ which have no contribution at all (§4.2). Further, as shown in Figure 3, 20%-50% of data

points has $p_A < 0.5$ after training converges on CIFAR-10. Therefore, we propose to discard hard samples directly during training, so that they explicitly have diminished effect on the final training convergence. Specifically, given a warm-up epoch $E_t$ and a confidence threshold $p_t$, we discard all data samples with $p_A < p_t$ at epoch $E_t$. We fix the number of steps taken by gradient descent and re-iterate on the distilled dataset when necessary to minimize the difference in training budget. After the discard, Gaussian training also ignores hard inputs, similar to SOTA algorithms.

### 5.2. Data Reweighting with Certified Radius

ACR relates non-linearly to $p_A$, and the growth of the certified radius is much faster when improving easy samples. We account for this by reweighing the data points based on their certified radius. Specifically, we use the approximate (normalized) certified radius as the weight of the data point, realized by controlling the sampling frequency of each data point. The weight $w$ of every data point $x$ is defined as:

$$\hat{p}_A = \text{LowerConfBound}(C, N, 1 - \alpha)$$
$$w = \max(1, \Phi^{-1}(\hat{p}_A)/\Phi^{-1}(p_{\min})),$$

where $C$ is the count of correctly classified noisy samples and $p_{\min}$ is the reference probability threshold. Note that the estimation of $\hat{p}_A$ aligns with the certification. We clip the weight at 1 to avoid zero weights. To minimize computational overhead, we evaluate $\hat{p}_A$ every 10 epoch with $N = 16$ and $\alpha = 0.1$ throughout the paper. We set $p_{\min} = 0.75$ because this is the minimum probability that has positive certified radius under this setting. The sampling weight curve is visualized in Figure 4. After the reweight, ACR has an approximately linear connection to $p_A$ when $w > 1$, and easy samples are sampled more frequently than hard samples so that Gaussian training also improves $p_A$ further for easy samples.

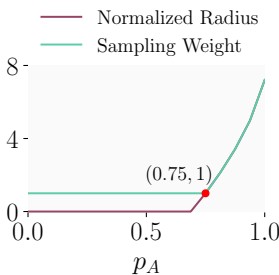

Figure 4: Certified radius (divided by the value at $p_A = 0.75$) and the sampling weight of the data against $p_A$.

---

**Algorithm 1** Adaptive Attack

> **function** ADAPTIVEADV($f, x, c, \boldsymbol{\delta}, T, \epsilon$)
> $\boldsymbol{\delta}^* \leftarrow \boldsymbol{\delta}$
> **for** $t = 1$ to $T$ **do**
>    **if** $f(x + \boldsymbol{\delta}^*) \neq c$ **then**
>      break
>    **end if**
>    $\boldsymbol{\delta}^* \leftarrow$ one step PGD attack on $\boldsymbol{\delta}^*$ with step size $\epsilon$
>    $\boldsymbol{\delta}^* \leftarrow \|\boldsymbol{\delta}\|_2 \cdot \boldsymbol{\delta}^* / \|\boldsymbol{\delta}^*\|_2$
> **end for**
> **return** $\boldsymbol{\delta}^*$
> **end function**

---

### 5.3. Adaptive Attack on the Sphere

SOTA algorithms re-balance the gradient norm of easy and hard samples in contrast to Gaussian training (Table 2). In addition, when $p_A$ is close to 1, Gaussian training can hardly find a useful noise sample to improve $p_A$ further. To tackle this issue, we propose to apply adaptive attack on the noise samples to balance samples with different $p_A$. Specifically, we use Projected Gradient Descent (PGD) (Madry et al., 2018) to find the nearest noise to the Gaussian noise which can make the base classifier misclassify. Formally, we construct

$$\boldsymbol{\delta}^* = \underset{f(x+\boldsymbol{\delta}) \neq c, \|\boldsymbol{\delta}\|_2 = \|\boldsymbol{\delta}_0\|_2}{\arg\min} \|\boldsymbol{\delta} - \boldsymbol{\delta}_0\|_2,$$

where $\boldsymbol{\delta}_0$ is a random Gaussian noise sample. Note that when $x + \boldsymbol{\delta}_0$ makes the base classifier misclassify, we have $\boldsymbol{\delta}^* = \boldsymbol{\delta}_0$, thus hard inputs are not affected by the adaptive attack. In addition, we remark that we do not constrain $\boldsymbol{\delta}^*$ to be in the neighborhood of $\boldsymbol{\delta}_0$ which is adopted by CAT-RS (Jeong et al., 2023); instead, we only maintain the $l_2$ norm of the noise, thus allowing the attack to explore a much larger space. This is because for every $\boldsymbol{\delta}^*$ such that $\|\boldsymbol{\delta}^*\|_2 = \|\boldsymbol{\delta}_0\|_2$, the value of the probability density function at $\boldsymbol{\delta}^*$ is the same as $\boldsymbol{\delta}_0$. We formalize this fact in Theorem 2.

**Theorem 2.** Assume $\boldsymbol{\delta}_1, \boldsymbol{\delta}_2 \in \mathbb{R}^d$ and $\boldsymbol{\delta}_1 \neq \boldsymbol{\delta}_2$. If $\|\boldsymbol{\delta}_1\|_2 = \|\boldsymbol{\delta}_2\|_2$, then $p_{\mathcal{N}(0,\sigma^2 I_d)}(\boldsymbol{\delta}_1) = p_{\mathcal{N}(0,\sigma^2 I_d)}(\boldsymbol{\delta}_2)$ for

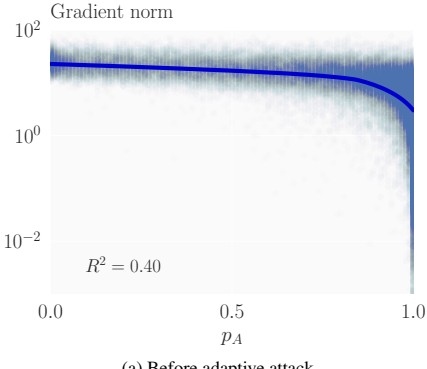

(a) Before adaptive attack

(b) After adaptive attack

Figure 5: Comparison of the gradient norm distributions for different $p_A$ before and after the adaptive attack, $\sigma = 0.5$. Note the log scale of y axis.

every $\sigma > 0$, where $p_{\mathcal{N}(0,\sigma^2 I_d)}$ is the probability density function of the isometric Gaussian distribution with mean 0 and covariance $\sigma^2 I_d$.

The proof of Theorem 2 is deferred to App. A.2. Figure 5 visualizes the gradient norm distributions for different $p_A$, before and after the adaptive attack. We observe that the adaptive attack balances the gradient norm of easy and hard samples. Before the attack, the gradient norm of easy samples is much smaller than that of hard samples, while after the attack, the gradient norm of easy samples is amplified without interfering the gradient norm of hard samples. Therefore, with the adaptive attack, Gaussian training obtains a similar gradient norm distribution to SOTA algorithms, and it can find effective noise samples more efficiently. Pseudocode of the adaptive attack is shown in Algorithm 1, and more detailed description is provided in App. C.3.

### 5.4. Overall Training Procedure

We now describe how the above three modifications are combined. At the beginning, we train the model with the Gaussian training (Cohen et al., 2019), which samples $m$ noisy points from the isometric Gaussian distribution uniformly at random and uses the average loss of noisy inputs as the training loss. When we reach the pre-defined warm-

up epoch $E_t$, all data points with $p_A < p_t$ are discarded, and the distilled dataset is used thereafter, as described in §5.1. After this, we apply dataset reweight and the adaptive attack to training. Specifically, every 10 epoch after $E_t$ (including $E_t$), we evaluate the model with the procedure described in §5.2 and assign the resulting sampling weight to each sample in the train set. In addition, we use the adaptive attack described in §5.3 to generate the noises used for training. The pseudocode is shown in Algorithm 2 in App. C.1.

# 6. Experimental Evaluation

We now evaluate our method proposed in §5 extensively. Overall, simply focusing on easy samples is enough to replicate the advances in RS training strategies, sometimes even surpassing existing SOTA algorithms, e.g., , on CIFAR-10. Experiment details, including hyperparameters, are provided in App. C.1.

**Baselines.** We compare our method to the following methods: Gaussian (Cohen et al., 2019), SmoothAdv (Salman et al., 2019), MACER (Zhai et al., 2020), Consistency (Jeong & Shin, 2020), SmoothMix (Jeong et al., 2021), and CAT-RS (Jeong et al., 2023). We always use the trained models provided by the authors if they are available and otherwise reproduce the results with the same setting as the original paper. For CIFAR-10, we set $m = 4$ for Gaussian training and our method, since this is the standard setting for SOTA methods (Jeong et al., 2023).

**Main Result.** Table 3 shows the ACR of different methods on CIFAR-10. We further include the results of independent runs in App. E.1 to validate the significance of our results. Our method consistently outperforms all baselines on ACR, which confirms the effectiveness of our modification to Gaussian training in increasing ACR. Further, our method successfully increases the certified accuracy at large radius, as we explicitly focus on easy inputs which is implicitly taken by other SOTA methods. Figure 6 further visualizes certified accuracy at different radii, showing that at small radii, SOTA methods (including ours) are systematically worse than Gaussian training, while at large radii, these methods consistently outperform Gaussian training.

Table 4 shows the results on IMAGENET, which is organized in the same way as Table 3. Our method replicates a large portion of the ACR advances. Concretely, for $\sigma = 0.25$, the proposed method recovers (0.529 - 0.476) / (0.532 - 0.476) = 94.6% of the advance; for $\sigma = 0.5$, it recovers (0.842 - 0.733) / (0.846 - 0.733) = 96.5%; for $\sigma = 1.0$, it recovers (1.042 - 0.875) / (1.071 - 0.875) = 85.2%. This confirms that our results generalize across datasets.

The success of our simple and intuitive modification to

Gaussian training suggests that existing approaches do not fully converge on strategies exploiting the weakness of ACR, and that the field should re-evaluate RS training strategies with better metrics instead of discarding them completely.

**Ablation Study.** We present a thorough ablation study in Table 5. When applied alone, all three components of our method improve ACR compared to Gaussian training. Combing two components arbitrarily improves the ACR compared to using only one component, and the best ACR is achieved when all three components are combined. This confirms that each component contributes to the improvement of ACR. In addition, they mostly improve the certified accuracy at large radii and reduce certified accuracy at small radii, which is consistent with our intuition that focusing on easy inputs can improve the ACR. More ablation on the hyperparameters is provided in App. D.

# 7. Beyond the ACR Metric

This work rigorously shows that the weakness of the main evaluation metric ACR has introduced strong biases into the development of RS training strategies. Therefore, the field has to seek better alternative metrics to evaluate the robustness of models under RS. When the certification setting (budget $N$ and confidence $\alpha$) is fixed, we suggest using the best certified accuracy at various radii. Specifically, models certified with smaller $\sigma$ tend to achieve higher certified accuracy at smaller radii, whereas models certified with larger $\sigma$ usually have higher certified accuracy at larger radii. To extensively illustrate the power of the developed training strategies and avoid such tradeoff and dependence on $\sigma$, we suggest reporting the best certified accuracy across $\sigma$ at each radius. In other words, this represents the setting where one first fixes an interested radius, and then try to develop the best model with the highest certified accuracy at the pre-defined radius. Notably, several studies already adopt this strategy (Carlini et al., 2023; Xiao et al., 2022), avoiding the weakness of ACR unintentionally. This evaluation is also consistent to the practice in deterministic certified training (Gowal et al., 2018; Mirman et al., 2018; Shi et al., 2021; Müller et al., 2023; Mao et al., 2023; 2024a;b; De Palma et al.; Balauca et al., 2024). Furthermore, the community should also consider RS methods with $\sigma$ as a variable rather than a hyperparameter, thus effectively removing the dependence on $\sigma$. While there are some preliminary works in this direction (Alfarra et al., 2022; Wang et al., 2021), they usually break the theoretical rigor (Súkeník et al., 2022). Rencent works propose sound methods to certify with various $\sigma$ (Súkeník et al., 2022; Jeong & Shin, 2024). However, room for improvement remains, and further research is needed.

When the certification setting varies, which is common in

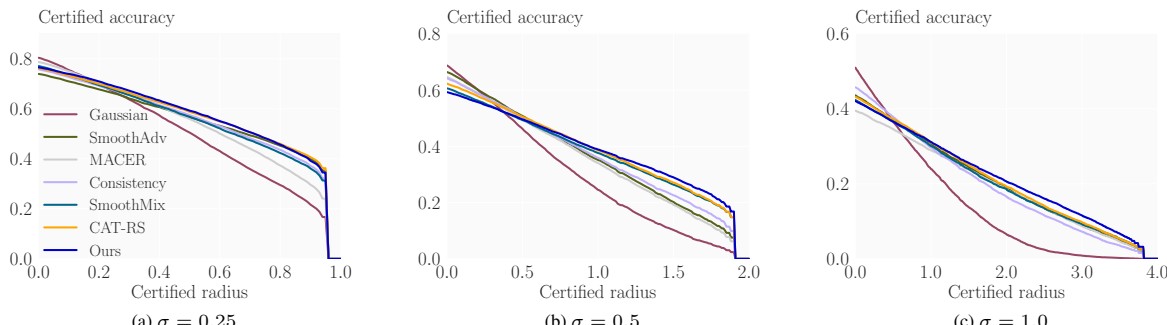

Figure 6: Certified radius-accuracy curve on CIFAR-10 for different methods.

| $\sigma$ | Methods | ACR | 0.00 | 0.25 | 0.50 | 0.75 | 1.00 | 1.25 | 1.50 | 1.75 | 2.00 | 2.25 | 2.50 |
|---|---|---|---|---|---|---|---|---|---|---|---|---|---|
| | Gaussian | 0.486 | **81.3** | 66.7 | 50.0 | 32.4 | 0.0 | 0.0 | 0.0 | 0.0 | 0.0 | 0.0 | 0.0 |
| | MACER | 0.529 | 78.7 | 68.3 | 55.9 | 40.8 | 0.0 | 0.0 | 0.0 | 0.0 | 0.0 | 0.0 | 0.0 |
| | SmoothAdv | 0.544 | 73.4 | 65.6 | 57.0 | 47.5 | 0.0 | 0.0 | 0.0 | 0.0 | 0.0 | 0.0 | 0.0 |
| 0.25 | Consistency | 0.547 | 75.8 | 67.4 | 57.5 | 46.0 | 0.0 | 0.0 | 0.0 | 0.0 | 0.0 | 0.0 | 0.0 |
| | SmoothMix | 0.543 | 77.1 | 67.6 | 56.8 | 45.0 | 0.0 | 0.0 | 0.0 | 0.0 | 0.0 | 0.0 | 0.0 |
| | CAT-RS | 0.562 | 76.3 | 68.1 | 58.8 | 48.2 | 0.0 | 0.0 | 0.0 | 0.0 | 0.0 | 0.0 | 0.0 |
| | **Ours** | **0.564** | 76.6 | **69.1** | **59.3** | **48.3** | 0.0 | 0.0 | 0.0 | 0.0 | 0.0 | 0.0 | 0.0 |
| | Gaussian | 0.562 | **68.7** | 57.6 | 45.7 | 34.0 | 23.7 | 15.9 | 9.4 | 4.8 | 0.0 | 0.0 | 0.0 |
| | MACER | 0.680 | 64.7 | 57.4 | 49.5 | 42.1 | 34.0 | 26.4 | 19.2 | 12.0 | 0.0 | 0.0 | 0.0 |
| | SmoothAdv | 0.684 | 65.3 | **57.8** | 49.9 | 41.7 | 33.7 | 26.0 | 19.5 | 12.9 | 0.0 | 0.0 | 0.0 |
| 0.5 | Consistency | 0.716 | 64.1 | 57.6 | 50.3 | 42.9 | 35.9 | 29.1 | 22.6 | 16.0 | 0.0 | 0.0 | 0.0 |
| | SmoothMix | 0.738 | 60.6 | 55.2 | 49.3 | 43.3 | 37.6 | 32.1 | 26.4 | 20.5 | 0.0 | 0.0 | 0.0 |
| | CAT-RS | 0.757 | 62.3 | 56.8 | **50.5** | **44.6** | 38.5 | 32.7 | 27.1 | 20.6 | 0.0 | 0.0 | 0.0 |
| | **Ours** | **0.760** | 59.3 | 54.8 | 49.6 | 44.4 | **38.9** | **34.1** | **29.0** | **23.0** | 0.0 | 0.0 | 0.0 |
| | Gaussian | 0.534 | **51.5** | **44.1** | 36.5 | 29.4 | 23.8 | 18.2 | 13.1 | 9.2 | 6.0 | 3.8 | 2.3 |
| | MACER | 0.760 | 39.5 | 36.9 | 34.6 | 31.7 | 28.9 | 26.4 | 23.8 | 21.1 | 18.6 | 16.0 | 13.8 |
| | SmoothAdv | 0.790 | 43.7 | 40.3 | 36.9 | 33.8 | 30.5 | 27.0 | 24.0 | 21.4 | 18.4 | 15.9 | 13.4 |
| 1.0 | Consistency | 0.757 | 45.7 | 42.0 | **37.8** | 33.7 | 30.0 | 26.3 | 22.9 | 19.6 | 16.6 | 13.9 | 11.6 |
| | SmoothMix | 0.788 | 42.4 | 39.4 | 36.7 | 33.4 | 30.0 | 26.8 | 23.9 | 20.8 | 18.6 | 15.9 | 13.6 |
| | CAT-RS | 0.815 | 43.2 | 40.2 | 37.2 | **34.3** | 31.0 | 28.1 | 24.9 | 22.0 | 19.3 | 16.8 | 14.2 |
| | **Ours** | **0.844** | 42.0 | 39.4 | 36.5 | 33.9 | 31.1 | 28.4 | 25.6 | 23.1 | 20.6 | 18.3 | 16.1 |

Table 3: Comparison of certified test accuracy (%) at different radii and ACR on CIFAR-10. The best and the second best results are highlighted in bold and underline, respectively; for certified accuracy, we highlight those that are worse than Gaussian training at the same radius in gray.

| $\sigma$ | Methods | ACR | 0.0 | 0.5 | 1.0 | 1.5 | 2.0 | 2.5 | 3.0 | 3.5 |
|---|---|---|---|---|---|---|---|---|---|---|
| | Gaussian | 0.476 | **66.7** | 49.4 | 0.0 | 0.0 | 0.0 | 0.0 | 0.0 | 0.0 |
| 0.25 | SmoothAdv | **0.532** | 65.3 | 55.5 | 0.0 | 0.0 | 0.0 | 0.0 | 0.0 | 0.0 |
| | **Ours** | 0.529 | 64.2 | **55.8** | 0.0 | 0.0 | 0.0 | 0.0 | 0.0 | 0.0 |
| | Gaussian | 0.733 | **57.2** | 45.8 | 37.2 | 28.6 | 0.0 | 0.0 | 0.0 | 0.0 |
| | SmoothAdv | 0.824 | 53.6 | 49.4 | 43.3 | 36.8 | 0.0 | 0.0 | 0.0 | 0.0 |
| 0.5 | Consistency | 0.822 | 55.0 | **50.2** | **44.0** | 34.8 | 0.0 | 0.0 | 0.0 | 0.0 |
| | SmoothMix | **0.846** | 54.6 | 50.0 | 43.4 | **37.8** | 0.0 | 0.0 | 0.0 | 0.0 |
| | **Ours** | 0.842 | 55.5 | 48.6 | 43.8 | 37.6 | 0.0 | 0.0 | 0.0 | 0.0 |
| | Gaussian | 0.875 | **43.6** | 37.8 | 32.6 | 26.0 | 19.4 | 14.8 | 12.2 | 9.0 |
| | SmoothAdv | 1.040 | 40.3 | 37.0 | 34.0 | 30.0 | 26.9 | 24.6 | 19.7 | 15.2 |
| | Consistency | 0.982 | 41.6 | 37.4 | 32.6 | 28.0 | 24.2 | 21.0 | 17.4 | 14.2 |
| 1.0 | SmoothMix | 1.047 | 39.6 | 37.2 | 33.6 | 30.4 | 26.2 | 24.0 | 20.4 | 17.0 |
| | CAT-RS | **1.071** | 43.6 | **38.2** | **35.2** | 30.8 | 26.8 | 24.0 | 20.2 | 17.0 |
| | **Ours** | 1.042 | 41.0 | 37.4 | 33.6 | **31.0** | **27.0** | 23.2 | 19.4 | 15.8 |

Table 4: Comparison of certified test accuracy (%) at different radii and ACR on IMAGENET. This table is structured in the same way as Table 3.

| $\sigma$ | discard | dataset weight | adverserial | ACR | 0.00 | 0.25 | 0.50 | 0.75 | 1.00 | 1.25 | 1.50 | 1.75 | 2.00 | 2.25 | 2.50 |
|---|---|---|---|---|---|---|---|---|---|---|---|---|---|---|---|
| | | Gaussian | | 0.486 | 81.3 | 66.7 | 50.0 | 32.4 | 0.0 | 0.0 | 0.0 | 0.0 | 0.0 | 0.0 | 0.0 |
| | ✓ | | | 0.515 | 81.2 | 69.3 | 53.7 | 36.8 | 0.0 | 0.0 | 0.0 | 0.0 | 0.0 | 0.0 | 0.0 |
| | | ✓ | | 0.512 | 81.3 | 69.4 | 53.3 | 36.3 | 0.0 | 0.0 | 0.0 | 0.0 | 0.0 | 0.0 | 0.0 |
| 0.25 | | | ✓ | 0.537 | 76.7 | 66.7 | 55.6 | 44.3 | 0.0 | 0.0 | 0.0 | 0.0 | 0.0 | 0.0 | 0.0 |
| | ✓ | ✓ | | 0.523 | 81.1 | 69.7 | 54.6 | 38.3 | 0.0 | 0.0 | 0.0 | 0.0 | 0.0 | 0.0 | 0.0 |
| | ✓ | | ✓ | 0.550 | 77.4 | 68.5 | 57.7 | 45.4 | 0.0 | 0.0 | 0.0 | 0.0 | 0.0 | 0.0 | 0.0 |
| | | ✓ | ✓ | 0.554 | 75.0 | 67.1 | 58.1 | 48.1 | 0.0 | 0.0 | 0.0 | 0.0 | 0.0 | 0.0 | 0.0 |
| | ✓ | ✓ | ✓ | **0.564** | 76.6 | 69.1 | 59.3 | 48.3 | 0.0 | 0.0 | 0.0 | 0.0 | 0.0 | 0.0 | 0.0 |
| | | Gaussian | | 0.525 | 65.7 | 54.9 | 42.8 | 32.5 | 22.0 | 14.1 | 8.3 | 3.9 | 0.0 | 0.0 | 0.0 |
| | ✓ | | | 0.627 | 68.4 | 59.4 | 49.5 | 39.4 | 29.0 | 20.5 | 13.0 | 7.0 | 0.0 | 0.0 | 0.0 |
| | | ✓ | | 0.662 | 68.1 | 59.7 | 50.3 | 41.1 | 31.7 | 23.7 | 16.2 | 9.2 | 0.0 | 0.0 | 0.0 |
| 0.5 | | | ✓ | 0.701 | 63.4 | 56.2 | 49.1 | 41.7 | 34.5 | 28.2 | 22.1 | 16.5 | 0.0 | 0.0 | 0.0 |
| | ✓ | ✓ | | 0.672 | 68.5 | 60.1 | 51.0 | 41.8 | 32.4 | 24.2 | 16.7 | 9.3 | 0.0 | 0.0 | 0.0 |
| | ✓ | | ✓ | 0.731 | 63.4 | 56.8 | 50.1 | 43.7 | 37.0 | 30.8 | 24.4 | 18.0 | 0.0 | 0.0 | 0.0 |
| | | ✓ | ✓ | 0.741 | 56.1 | 52.1 | 47.3 | 43.2 | 38.6 | 34.1 | 29.1 | 23.1 | 0.0 | 0.0 | 0.0 |
| | ✓ | ✓ | ✓ | **0.760** | 59.3 | 54.8 | 49.6 | 44.4 | 38.9 | 34.1 | 29.0 | 23.0 | 0.0 | 0.0 | 0.0 |
| | | Gaussian | | 0.534 | 51.5 | 44.1 | 36.5 | 29.4 | 23.8 | 18.2 | 13.1 | 9.2 | 6.0 | 3.8 | 2.3 |
| | ✓ | | | 0.665 | 46.8 | 42.1 | 37.6 | 33.1 | 28.7 | 24.3 | 20.2 | 16.1 | 12.6 | 9.8 | 7.4 |
| | | ✓ | | 0.695 | 49.9 | 44.9 | 39.7 | 34.9 | 29.8 | 25.3 | 21.4 | 17.5 | 13.6 | 10.1 | 7.1 |
| 1.0 | | | ✓ | 0.690 | 47.3 | 42.0 | 37.0 | 32.0 | 27.1 | 23.2 | 19.9 | 16.6 | 13.6 | 10.8 | 8.4 |
| | ✓ | ✓ | | 0.736 | 48.8 | 44.5 | 40.3 | 35.9 | 31.4 | 27.3 | 22.9 | 19.1 | 15.2 | 11.7 | 8.7 |
| | ✓ | | ✓ | 0.771 | 47.0 | 43.1 | 39.1 | 35.0 | 30.9 | 27.1 | 23.6 | 20.2 | 17.0 | 14.1 | 11.6 |
| | | ✓ | ✓ | 0.818 | 39.7 | 37.5 | 34.8 | 32.5 | 29.9 | 27.7 | 25.3 | 22.6 | 20.1 | 17.9 | 15.8 |
| | ✓ | ✓ | ✓ | **0.844** | 42.0 | 39.4 | 36.5 | 33.9 | 31.1 | 28.4 | 25.6 | 23.1 | 20.6 | 18.3 | 16.1 |

Table 5: Ablation study on each component in our method.

applying RS to practice, metrics are required to reflect the power of RS training strategies under different certification budgets. To this end, we suggest reporting the empirical distribution of $p_A$ for a large $N$, e.g., $N = 10^5$ as commonly adopted. This approximates the true distribution of $p_A$ nicely when $N$ is large, and a higher $p_A$ usually leads to a larger certified radius consistently under different certification budgets. To this end, as long as one model has larger $P(p_A \geq p)$ for every $p \in [0.5, 1]$ than another model, we can claim the former is better consistently in different certification settings. A case study is included in App. B to further demonstrate the usage and properties of the empirical distribution of $p_A$.

While our modifications presented in §5 are not designed to improve robustness for the general data distribution, they show effectiveness in increasing certified accuracy at large radius. Other existing algorithms show similar effects. Therefore, it is important to note that the field has indeed made progress over the years. However, new metrics which consider robustness more uniformly should be developed to evaluate RS, and algorithms that outperform at every radius are encouraged. We hope this work can inspire future research in this direction.

## 8. Conclusion

This work rigorously demonstrates that Average Certified Radius (ACR) is a poor metric for Randomized Smoothing (RS). Theoretically, we prove that ACR of a trivial classifier can be arbitrarily large, and that an improvement on easy inputs contributes much more to ACR than the same amount of improvement on hard inputs. In addition, the comparison results based on ACR varies when the certification budget changes, thus unsuitable for comparing algorithms. Empirically, we show that all state-of-the-art (SOTA) strategies reduce the accuracy on hard inputs and only focus on easy inputs to increase ACR. Based on these novel insights, we develop strategies to amplify Gaussian training by focusing only on easy data points during training. Specifically, we discard hard inputs during training, reweigh the dataset with their contribution to ACR, and apply extreme optimization for easy inputs via adversarial noise selection. With these intuitive modifications to the simple Gaussian training, we replicate the effect of SOTA training algorithms and achieve the SOTA ACR. Overall, our results suggest the need for evaluating RS training with better metrics. To this end, we suggest reporting the empirical distribution of $p_A$ for a large $N$ in replacement of ACR. We hope this work will inspire future research in this direction and motivate the community to permanently abandon ACR in the evaluation of randomized smoothing.

## Reproducibility Statement

Experimental details, including hyperparameters, datasets, and training procedures, are provided in App. C. Result significance is confirmed in App. E.1. Code and models are available at `https://github.com/eth-sri/acr-weakness`.

# Acknowledgements

This work has been done as part of the EU grant ELSA (European Lighthouse on Secure and Safe AI, grant agreement no. 101070617) and the SERI grant SAFEAI (Certified Safe, Fair and Robust Artificial Intelligence, contract no. MB22.00088). Views and opinions expressed are however those of the authors only and do not necessarily reflect those of the European Union or European Commission. Neither the European Union nor the European Commission can be held responsible for them.

The work has received funding from the Swiss State Secretariat for Education, Research and Innovation (SERI).

# Impact Statement

This paper presents work whose goal is to advance the field of Machine Learning. There are many potential societal consequences of our work, none of which we feel must be specifically highlighted here.

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

# A. Deferred Proofs

## A.1. Proof of Theorem 1

Now we prove Theorem 1, restated here for convenience.

**Theorem 1.** For every $M > 0$ and $\alpha > 0$, there exists a trivial classifier $f$ which always predicts the same class with a certification budget $N > 0$, such that the ACR of $f$ is greater than $M$ with confidence at least $1 - \alpha$.

*Proof.* Assume we are considering a $K$-class classification problem with a dataset containing $T$ samples. Let $c^*$ be the most likely class and $X^*$ be the set of all samples with label $c^*$; then there are at least $\lceil T/K \rceil$ samples in $X^*$ due to the pigeonhole theorem. We then show that a trivial classifier $f$ which always predicts $c^*$ can achieve an ACR greater than $M$ with confidence at least $1 - \alpha$ with a proper budget $N$.

Note that $p_A = 1$ for $x \in X^*$, and thus the certified radius of $x \in X^*$ is $R(x) = \sigma \Phi^{-1}(\alpha^{1/N})$. Therefore, ACR $= \frac{1}{T} \left[ \sum_{x \in X^*} R(x) + \sum_{x \notin X^*} R(x) \right] \geq \frac{1}{T} \sum_{x \in X^*} R(x) \geq \frac{1}{K} \sigma \Phi^{-1}(\alpha^{1/N})$. Setting $N = \lceil \frac{\log(\Phi(MK/\sigma))}{\log(\alpha)} \rceil + 1$, we have ACR $> M$. $\qquad\square$

## A.2. Proof of Theorem 2

Now we prove Theorem 2, restated here for convenience.

**Theorem 2.** Assume $\boldsymbol{\delta}_1, \boldsymbol{\delta}_2 \in \mathbb{R}^d$ and $\boldsymbol{\delta}_1 \neq \boldsymbol{\delta}_2$. If $\|\boldsymbol{\delta}_1\|_2 = \|\boldsymbol{\delta}_2\|_2$, then $p_{\mathcal{N}(0,\sigma^2 I_d)}(\boldsymbol{\delta}_1) = p_{\mathcal{N}(0,\sigma^2 I_d)}(\boldsymbol{\delta}_2)$ for every $\sigma > 0$, where $p_{\mathcal{N}(0,\sigma^2 I_d)}$ is the probability density function of the isometric Gaussian distribution with mean 0 and covariance $\sigma^2 I_d$.

*Proof.* Let $\boldsymbol{\delta} = [q_1, q_2, \ldots, q_d] \in \mathbb{R}^d$ be sampled from $\mathcal{N}(0, \sigma^2 I_d)$. Then we have

$$\mathbb{P}(\boldsymbol{\delta}) = \frac{1}{(2\pi)^{d/2}\sigma^d} \exp\left(-\frac{1}{2\sigma^2} \sum_{i=1}^{d} q_i^2\right)$$
$$= \frac{1}{(2\pi)^{d/2}\sigma^d} \exp\left(-\frac{1}{2\sigma^2} \|\boldsymbol{\delta}\|_2^2\right).$$

This concludes the proof. $\qquad\square$

# B. On the Empirical Distribution of $p_A$

In this section, we include a case study to further demonstrate the idea we presented in §7 about using the empirical distribution of $p_A$ to evaluate the RS model. We first show how to calculate the certified accuracy for every interested radius and certification budget efficiently, given the ECDF of $p_A$. Then, we show how to convert an existing certified accuracy-radius curve to the ECDF of $p_A$, thus the literature results can be easily reused. These features facilitate

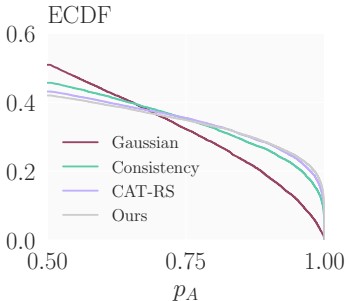

Figure 7: The empirical distribution of $p_A$ on CIFAR-10 with $\sigma = 1.0$ and $N = 10^5$. Note that this curve is independent of $\alpha$.

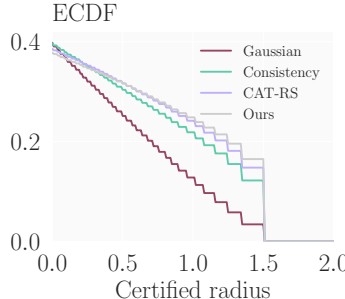

Figure 8: The certified accuracy-radius curve under $N = 100$ and $\alpha = 0.01$ on CIFAR-10 with $\sigma = 1.0$, converted from Figure 7.

practitioners to apply RS models in practice, as the certification budget usually varies in different scenarios. Finally, we re-evaluate previous RS training strategies with the ECDF of $p_A$.

## B.1. From ECDF to Certified Accuracy under Different Certification Settings

We consider four RS models (Gaussian, Consistency, CAT-RS and ours) trained on CIFAR-10. The ECDF of $p_A$ under $N = 10^5$ is shown in Figure 7. Assume now instead of certifying with $N = 10^5$ and $\alpha = 0.001$, we want to certify with $N = 100$ and $\alpha = 0.01$, which is less expensive and more practical to run on edge devices. We are interested in the certified accuracy-radius curve under this new certification setting.

Since the ECDF of $p_A$ is calculated with a large $N$, we assume it equals the true population of $p_A$. When it is computed with a relatively small $N$, the difference between ECDF and the true population might be non-trivial, and corrections should be done in such cases. We leave this as future work, and focus on the case where the ECDF is accurate enough to represent the true population.

We take the case where $r = 0.5$ to demonstrate the process. Given $r = 0.5$, $N = 100$ and $\alpha = 0.01$, we run a binary search on $p_A$ to identify the minimum $p_A$ required to certify the radius. In this case, the minimum $p_A$ is 0.81. Then, we look up the ECDF curve to find the cor-

responding $P(p_A \geq 0.81)$. For our method in Figure 7, $P(p_A \geq 0.81) = 0.316$. Therefore, the expected certified accuracy under the new certification setting is 0.316. We note that this is an expected value, as multiple runs can lead to different certified accuracies due to the randomness of the noise samples.

Now, repeat this process for all radii of interest, and we can obtain the certified accuracy-radius curve under the new certification setting. The results are shown in Figure 8. We remark that no inference of the model is conducted in this process, thus the overhead is negligible.

### B.2. From Certified Radius to ECDF

Computing ECDF for a given model requires the same computation as computing the certified accuracy-radius curve, as the former is a middle step of the latter. Fortunately, given the certified accuracy-radius curve commonly presented in the literature with large $N$, we can also convert it to ECDF of $p_A$.

The idea is to revert the process in App. B.1. Assume we are given the certified accuracy-radius curve when $N = 10^5$ and $\alpha = 0.001$. Given an interested $p_A$, we first calculate the maximum radius that can be certified with this $p_A$ under this certification setting. Then, we can look up the certified radius for this $p_A$ in the curve, and obtain the corresponding certified accuracy. This is the probability of $p_A$ being greater than the given $p_A$, ignoring the difference between ECDF and true population. Repeat this process for all interested $p_A$, and we can obtain the ECDF of $p_A$.

### B.3. Evaluating Current RS with ECDF

For future reference, we evaluate all existing methods, including Gaussian, Consistency, SmoothAdv, SmoothMix, MACER, CAT-RS and ours, using the ECDF of $p_A$, together with our proposed algorithm. We use the same dataset and model as in §6, i.e. ResNet-110 on CIFAR-10 with $\sigma \in \{0.25, 0.5, 1.0\}$ and $N = 10^5$. Whenever an official public model is available, we use it to compute the ECDF of $p_A$. Otherwise, we re-train the model with the same hyperparameters as in the original paper.

The ECDF of $p_A$ is shown in Figure 9. The related data point is available in our GitHub repository. We note that the ECDF is independent of $\alpha$, thus it can be used to evaluate the certified accuracy under any certification setting. Given the ECDF metric, model $\mathcal{A}$ is better than model $\mathcal{B}$ if and only if $\mathcal{A}$ has higher ECDF than $\mathcal{B}$ for all $p_A \geq 0.5$. Therefore, it is possible that neither of two models is better than the other.

For various $\sigma$, the Gaussian model is always the best in the low $p_A$ region, i.e. from $p_A = 0.5$ to around 0.65. Although the our model consistently has the highest ACR

| $\sigma$ | 0.25 | 0.5 | 1.0 |
|---|---|---|---|
| $m$ | 4 | 4 | 4 |
| $E_t$ | 60 | 70 | 60 |
| $p_t$ | 0.5 | 0.4 | 0.4 |
| $T$ | 3 | 6 | 4 |
| $\epsilon$ | 0.25 | 0.25 | 0.5 |

Table 6: Hyperparameters we use on CIFAR-10.

under all $\sigma$, it only outperforms other models in the high $p_A$ region. For example, with $\sigma = 0.5$, our model is the worst when $p_A$ is below 0.75. We list some cases where two models have strict orders: with $\sigma = 0.25$, the CAT-RS model is better than Consistency and SmoothAdv, and our model is better than CAT-RS; with $\sigma = 0.5$, the CAT-RS model is better than the SmoothMix model; with $\sigma = 1.0$, the CAT-RS model is better than SmoothMix and MACER, and our model is better than MACER.

## C. Experiment Details

### C.1. Training Details

We follow the standard training protocols in previous works. Specifically, we use ResNet-110 (He et al., 2016) on CIFAR-10 and ResNet-50 on IMAGENET, respectively. For CIFAR-10, we train ResNet-110 for 150 epochs with an initial learning rate of 0.1, which is decreased by a factor of 10 in every 50 epochs. The inital learning rate of training ResNet-50 on IMAGENET is 0.1 and it is decreased by a factor of 10 in every 30 epochs within 90 epochs. We investigate three noise levels, $\sigma = 0.25, 0.5, 1.0$. We use SGD with momentum 0.9. The following hyperparameter is tuned for each setting:

$m =$ the number of noise samples for each input

$E_t =$ the epoch to discard hard data samples

$p_t =$ the threshold of $p_A$ to discard hard data samples

$T =$ the maximum number of steps of the PGD attack

$\epsilon =$ the step size of the PGD attack

All other hyperparameter is fixed to the default values. Specifically, for CIFAR-10 we use 100 noise samples to calculate $p_A$ when discarding hard samples. When updating dataset weight, we use 16 noise samples to calculate $p_A$, with $p_{\min} = 0.75$. The hyperparameter tuned on CIFAR-10 is shown in Table 6. For IMAGENET, we skip the 'Data Reweighting with Certified Radius' step for faster training, and use a Gaussian-pretrained ResNet-50 to evaluate the $p_A$ with 50 noise samples. The hyperparameters tuned on IMAGENET is shown in Table 7.

In Algorithm 2, we present a pseudocode for the overall training procedure described in §5.

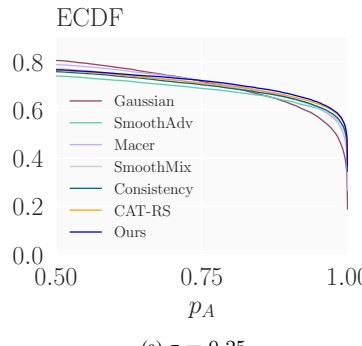

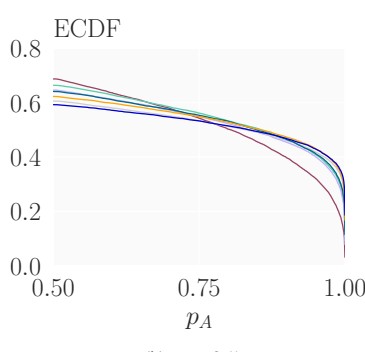

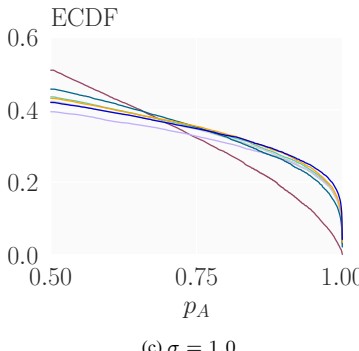

(a) $\sigma = 0.25$             (b) $\sigma = 0.5$             (c) $\sigma = 1.0$

Figure 9: The empirical distribution of $p_A$ on CIFAR-10 with different $\sigma$ and $N = 10^5$. This figure is an extension of Figure 7 on more $\sigma$ and algorithms.

| $\sigma$ | 0.25 | 0.5 | 1.0 |
|---|---|---|---|
| $m$ | 1 | 1 | 2 |
| $E_t$ | 0 | 0 | 0 |
| $p_t$ | 0.2 | 0.2 | 0.1 |
| $T$ | 2 | 2 | 1 |
| $\epsilon$ | 0.5 | 1.0 | 2.0 |

Table 7: Hyperparameters we used on IMAGENET.

### C.2. Certification Algorithm

We use the CERTIFY function in (Cohen et al., 2019) to calculate the certified radius, same as previous baselines, where $N = 10^5$ and $\alpha = 0.001$.

### C.3. Adversarial Attack Algorithm

In the adaptive attack, we implement the $l_2$ PGD and $l_\infty$ PGD attack. The difference between them lies in how the noise is updated based on the gradient. For $l_2$ PGD, the noise is updated as follows:

$$\delta^* = \delta + \epsilon \cdot \nabla_\delta L(f(x + \delta), y) / \|\nabla_\delta L(f(x + \delta), y)\|_2,$$

while for $l_\infty$ PGD, the noise is updated as follows:

$$\delta^* = \delta + \epsilon \cdot \text{sign}(\nabla_\delta L(f(x + \delta), y)),$$

where $L$ is the loss function, $f$ is the model, $x$ is the input, $y$ is the label, $\delta$ is the noise, $\delta^*$ is the updated noise, sign is the sign function, and $\epsilon$ is the step size. Unless stated otherwise, our experiments are always conducted with the $l_2$ PGD attack. We show the results with the $l_\infty$ PGD attack on CIFAR-10 in App. E.4.

## D. Additional Ablation Studies

In this section, we provide additional ablation studies on the effect of different hyperparameters. All results are based on $\sigma = 0.5$ and CIFAR-10. Unless stated otherwise, all hyperparameters are the same as those in Table 6.

---

**Algorithm 2** Overall Training Procedure

**Input:** Train dataset $\mathcal{D}$, noise level $\sigma$, hyperparameters $E_t, p_t, m, T, \epsilon$
Initialize the model $f$
**for** epoch $= 1$ to $N_{\text{epoch}}$ **do**
  **if** epoch $< E_t$ **then**
    Sample $\delta_1, \ldots, \delta_m \sim \mathcal{N}(0, \sigma^2 I)$
    Perform Gaussian training with $\delta_1, \ldots, \delta_m$
    continue
  **else if** epoch $= E_t$ **then**
    Discard hard data samples in $\mathcal{D}$ with $p_A < p_t$ to form $\mathcal{D}'$
  **end if**
  **if** epoch $\%10 = 0$ **then**
    update dataset weight according to §5.2
  **end if**
  Sample $|\mathcal{D}|$ data samples from $\mathcal{D}'$ with replacement to form the train set $\mathcal{D}''$
  **for** input $x$, label $c$ in $\mathcal{D}''$ **do**
    Sample $\delta_1, \ldots, \delta_m \sim \mathcal{N}(0, \sigma^2 I)$
    **for** i $= 1$ to $m$ **do**
      $\delta_i^* \leftarrow \text{ADAPTIVEADV}(f, x, c, \delta_i, T, \epsilon)$
    **end for**
    Perform Gaussian training with $\delta_1^*, \ldots, \delta_m^*$
  **end for**
**end for**

---

### D.1. Effect of Discarding Hard Data

We investigate the effect of the two hyperparameters $E_t$ and $p_t$ in discarding hard samples. The results are shown in Table 9 and Table 10. If hard samples are discarded too early, it leads to a decrease in accuracy for all radii, as many potential easy samples are discarded before their $p_A$ reaches $p_t$. Conversely, discarding hard samples too late allows more samples to remain, resulting in improved clean accuracy but reduced performance on easy samples, as the training focuses less on them.

| $\sigma$ | ACR | 0.00 | 0.25 | 0.50 | 0.75 | 1.00 | 1.25 | 1.50 | 1.75 | 2.00 | 2.25 | 2.50 |
|---|---|---|---|---|---|---|---|---|---|---|---|---|
| 0.25 | 0.5613±0.0022 | 76.43±0.11 | 68.58±0.34 | 58.93±0.34 | 48.15±0.10 | 0.0 | 0.0 | 0.0 | 0.0 | 0.0 | 0.0 | 0.0 |
| 0.5 | 0.7587±0.0011 | 59.00±0.20 | 54.40±0.26 | 49.37±0.16 | 44.31±0.14 | 39.09±0.19 | 34.14±0.05 | 28.91±0.05 | 22.83±0.13 | 0.0 | 0.0 | 0.0 |
| 1.0 | 0.8434±0.0003 | 41.74±0.43 | 39.23±0.44 | 36.46±0.31 | 33.73±0.33 | 30.95±0.27 | 28.30±0.08 | 25.62±0.09 | 23.22±0.15 | 20.74±0.15 | 18.40±0.25 | 16.12±0.19 |

Table 8: Mean and Standard deviation of ACR and certified accuracy on CIFAR-10 using our method. The results are based on three independent runs with different random seeds.

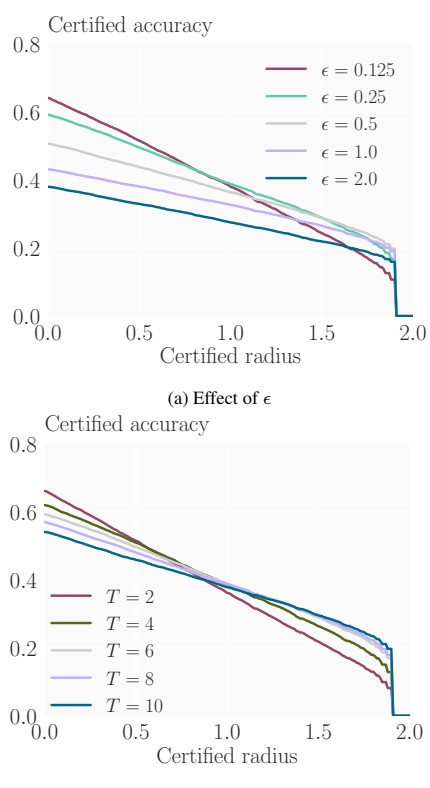

(a) Effect of $\epsilon$

(b) Effect of $T$

Figure 10: Certified radius-accuracy curves on CIFAR-10 for different $\epsilon$ and $T$.

### D.2. Effect of Adaptive Attack

For the adaptive attack, we evaluate the effect of the step size $\epsilon$ and the number of steps $T$. The results are shown in Table 11 and Table 12. In general, increasing the number of steps strengthens the attack, which helps find effective noise samples for extremely easy inputs. As a result, performance improves at larger radii, but relatively less attention is given to hard samples, leading to a decrease in clean accuracy. Similarly, increasing the step size in a moderate range ($\epsilon < 1.0$) has a similar effect. However, if the step size is too large, the attack loses effectiveness, resulting in decreased performance at all radii.

## E. Additional Results

### E.1. Result Significance

Table 8 shows the mean and standard deviation of the ACR and certified accuracy-radius curve on CIFAR-10 using our

| $E_t$ | ACR | 0.00 | 0.25 | 0.50 | 0.75 | 1.00 | 1.25 | 1.50 | 1.75 |
|---|---|---|---|---|---|---|---|---|---|
| 50 | 0.738 | 58.0 | 53.2 | 48.3 | 42.9 | 38.1 | 33.0 | 27.5 | 21.8 |
| 70 | 0.760 | 59.3 | 54.8 | 49.6 | 44.4 | 38.9 | 34.1 | 29.0 | 23.0 |
| 90 | 0.751 | 60.1 | 54.9 | 49.6 | 44.1 | 38.6 | 33.4 | 27.5 | 21.4 |
| 110 | 0.746 | 60.0 | 55.1 | 49.5 | 43.7 | 38.0 | 33.3 | 27.2 | 21.0 |
| 130 | 0.738 | 61.4 | 56.1 | 50.2 | 44.2 | 37.5 | 31.7 | 25.5 | 19.2 |

Table 9: Effect of the discarding epoch $E_t$.

| $p_t$ | ACR | 0.00 | 0.25 | 0.50 | 0.75 | 1.00 | 1.25 | 1.50 | 1.75 |
|---|---|---|---|---|---|---|---|---|---|
| 0.3 | 0.753 | 59.5 | 54.5 | 49.4 | 44.3 | 39.0 | 33.7 | 28.0 | 21.9 |
| 0.4 | 0.760 | 59.3 | 54.8 | 49.6 | 44.4 | 38.9 | 34.1 | 29.0 | 23.0 |
| 0.5 | 0.757 | 60.0 | 54.7 | 49.9 | 44.5 | 38.9 | 33.7 | 27.9 | 22.1 |
| 0.6 | 0.758 | 59.2 | 54.6 | 49.5 | 44.3 | 39.4 | 34.1 | 28.7 | 22.2 |
| 0.7 | 0.749 | 59.5 | 54.4 | 49.0 | 43.7 | 38.7 | 33.2 | 27.7 | 21.7 |
| 0.8 | 0.740 | 59.5 | 54.4 | 49.1 | 43.6 | 38.1 | 32.8 | 26.8 | 20.8 |

Table 10: Effect of the discarding $p_A$ threshold $p_t$.

best hyperparameters reported in Table 6. The results are based on three independent runs with different random seeds. The standard deviation is relatively small, confirming the stability of our results.

### E.2. Empirical Distribution of $p_A$ on IMAGENET

We extend Figure 3 to IMAGENET, shown as Figure 11. We observe that for SOTA methods, there are more samples with $p_A$ below 0.5 compared to Gaussian training, which is consistent to the results on CIFAR-10 in §4.3.

### E.3. Discarded Data Ratio

In this section, we are interested in how many data samples are discarded during training. The remaining data ratios under different $\sigma$ on CIFAR-10 are 91%, 80% and 58%, for $\sigma = 0.25$, 0.5 and 1.0, respectively. The results indicate that even though a significant portion of the training set is discarded during training, we still achieve a better ACR than the current SOTA model according to Table 3. Further, this confirms that larger $\sigma$ leads to more discarded data samples, which is consistent with the intuition.

### E.4. Replacing $l_2$ with $l_\infty$ PGD Attack

In this section, we replace the $L_2$ attack with $l_\infty$ PGD attack. The results are shown in Table 13 and Table 14. For $l_\infty$ attack, we use much smaller step sizes $\epsilon$ to have similar attack strengths to $l_2$ attack. Specifically, we investigate $\sqrt{d} \cdot \epsilon = 0.125, 0.25, 1.5, 1.0$ for $l_\infty$ attack, where $d$ is the

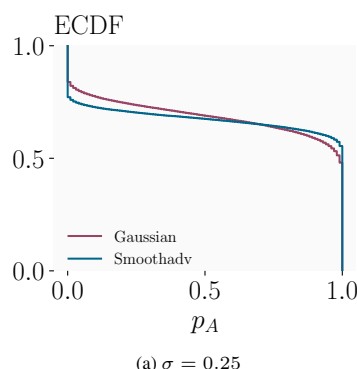
(a) $\sigma = 0.25$

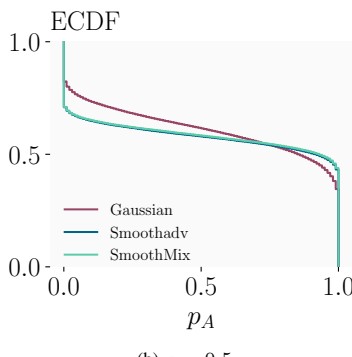
(b) $\sigma = 0.5$

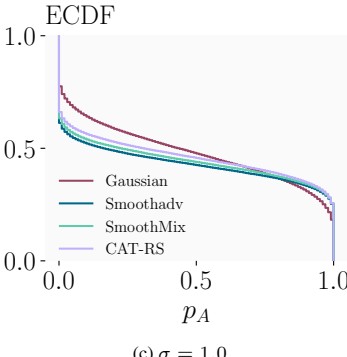
(c) $\sigma = 1.0$

Figure 11: The empirical cumulative distribution (ECDF) of $p_A$ on IMAGENET for models trained and certified with various $\sigma$ with different training algorithms.

| $\epsilon$ | ACR | 0.00 | 0.25 | 0.50 | 0.75 | 1.00 | 1.25 | 1.50 | 1.75 |
|---|---|---|---|---|---|---|---|---|---|
| 0.125 | 0.744 | 64.2 | 58.1 | 51.6 | 44.7 | 38.1 | 31.4 | 24.4 | 17.1 |
| 0.250 | 0.760 | 59.3 | 54.8 | 49.6 | 44.4 | 38.9 | 34.1 | 29.0 | 23.0 |
| 0.500 | 0.703 | 50.8 | 47.4 | 43.9 | 40.3 | 36.4 | 32.9 | 28.8 | 24.4 |
| 1.000 | 0.624 | 43.2 | 40.7 | 38.2 | 35.4 | 32.7 | 29.7 | 26.6 | 22.7 |
| 2.000 | 0.533 | 38.1 | 35.6 | 33.0 | 30.5 | 27.6 | 24.9 | 22.0 | 19.0 |

Table 11: Effect of the adversarial attack step size $\epsilon$.

| $T$ | ACR | 0.00 | 0.25 | 0.50 | 0.75 | 1.00 | 1.25 | 1.50 | 1.75 |
|---|---|---|---|---|---|---|---|---|---|
| 2 | 0.719 | 66.2 | 58.9 | 51.5 | 43.7 | 36.2 | 28.9 | 21.8 | 14.5 |
| 4 | 0.752 | 62.0 | 56.7 | 50.9 | 44.6 | 38.6 | 32.7 | 26.3 | 19.7 |
| 6 | 0.760 | 59.3 | 54.8 | 49.6 | 44.4 | 38.9 | 34.1 | 29.0 | 23.0 |
| 8 | 0.748 | 57.0 | 52.6 | 48.0 | 43.4 | 38.7 | 34.2 | 29.2 | 23.6 |
| 10 | 0.732 | 54.1 | 49.9 | 45.9 | 42.0 | 37.8 | 34.0 | 29.5 | 24.6 |

Table 12: Effect of the number of steps of adversarial attack $T$.

dimension of the input. After the thorough experiments, we find that using $\sqrt{d} \cdot \epsilon = 0.25$ with other hyperparameters fixed to the values in Table 6 has the best ACR. The $l_\infty$ attack has a similar effect to the $l_2$ attack. Increasing the step size and the number of steps of the attack result in better accuracy at large radii but lower accuracy at small radii.

| $\sqrt{d} \cdot \epsilon$ | ACR | 0.00 | 0.25 | 0.50 | 0.75 | 1.00 | 1.25 | 1.50 | 1.75 |
|---|---|---|---|---|---|---|---|---|---|
| 0.125 | 0.722 | 66.3 | 59.6 | 51.8 | 44.3 | 36.4 | 29.1 | 21.3 | 13.7 |
| 0.250 | 0.746 | 62.7 | 57.0 | 51.1 | 44.8 | 38.4 | 32.0 | 25.4 | 18.5 |
| 0.500 | 0.727 | 55.7 | 51.3 | 46.9 | 42.5 | 37.7 | 32.8 | 28.0 | 22.6 |
| 1.000 | 0.640 | 46.4 | 43.3 | 39.9 | 36.7 | 33.5 | 30.0 | 25.9 | 22.0 |

Table 13: ACR and certified accuracy under the $l_\infty$ PGD attack with $\sigma = 0.5$ on CIFAR-10 with different step size.

| $T$ | ACR | 0.00 | 0.25 | 0.50 | 0.75 | 1.00 | 1.25 | 1.50 | 1.75 |
|---|---|---|---|---|---|---|---|---|---|
| 3 | 0.718 | 66.0 | 59.0 | 51.6 | 43.8 | 36.2 | 28.8 | 21.2 | 14.0 |
| 4 | 0.731 | 65.0 | 58.5 | 51.5 | 44.2 | 37.3 | 29.9 | 23.1 | 15.8 |
| 5 | 0.736 | 64.2 | 57.8 | 51.4 | 44.6 | 37.7 | 30.6 | 23.7 | 16.5 |
| 6 | 0.746 | 62.7 | 57.0 | 51.1 | 44.8 | 38.4 | 32.0 | 25.4 | 18.5 |
| 7 | 0.743 | 61.5 | 56.4 | 50.6 | 44.4 | 38.1 | 32.4 | 25.7 | 19.1 |
| 8 | 0.742 | 60.1 | 54.8 | 49.7 | 43.7 | 38.0 | 32.7 | 26.9 | 20.3 |

Table 14: ACR and certified accuracy under the $l_\infty$ PGD attack with $\sigma = 0.5$ on CIFAR-10 with different number of steps.

