# OpenReview forum: "Average Certified Radius is a Poor Metric for Randomized Smoothing"
_ICML.cc/2025/Conference — ICML 2025 poster_

### Official Review · Reviewer_gJ9M · 2025-03-05

**Overall Recommendation:** 2

**Summary:**

This paper studies the shortcomings of Aversage Certified Radius (ACR) as a performance metric for randomized smoothing. It shows theoretically that this metric can be “hacked” by a trivial classifier with an arbitrarily large certified radius on a small number of “easy” input points, thereby achieving SOTA performance under this metric. It confirms this theoretical finding empirically by designing model training strategies that discount hard input samples, prioritize easy samples, and put more weight on higher certified radii. Based on its findings, it argues for the discontinuation of ACR as a metric for evaluating randomized smoothing-based robustness techniques.

**Claims And Evidence:**

1. The paper argues that ACR is not a proper metric for evaluating randomized smoothing. However, most works in this field do not use ACR as the sole evaluation metric. The primary evaluation metric is certified accuracy at different values of the certified radius. This metric does not have the shortcomings of the ACR metric. Even the two works cited by this paper to argue that ACR is being used by the community continuously only use ACR as a secondary evaluation metric together with certified accuracy. Thus, the paper’s claim that ACR “has emerged as the most important metric for comparing methods and tracking progress in the field” is debatable.

2. The theoretical claims regarding a trivial classifier achieving infinite ACR by over-optimizing on easy samples have been validated with short proofs of correctness. However, the theoretical claims are straightforward and not very surprising. Please see my comments under “Theoretical Claims”.

**Essential References Not Discussed:**

Several well-known works in this field do not use ACR as their evaluation metric. For instance, [1] and [2] evaluate the certified robustness of RL agents using metrics such as certified reward, which does not suffer from the weaknesses of ACR. The paper should include more examples of works in the field and discuss the metrics used by them to give the reader a better understanding of how commonly the ACR metric is used as the primary evaluation metric in the literature.

[1] CROP: Certifying Robust Policies for Reinforcement Learning through Functional Smoothing, Wu et al, ICLR 2022.

[2] Policy Smoothing for Provably Robust Reinforcement Learning, Kumar et al, ICLR 2022.

**Experimental Designs Or Analyses:**

The experiments and analyses are reasonable for showing the weaknesses of the ACR metric.

**Methods And Evaluation Criteria:**

The paper proposes three modifications to training with Gaussian noise, namely discarding hard inputs, reweighting samples with approximate certified radius and attacking the noised samples, to develop a new method to achieve state-of-the-art performance under the ACR metric. This method shows that improving the ACR metric is indeed possible without actually making the model more robust. However, the method only achieves marginal improvements in ACR over existing methods, as shown in Table 2.

**Other Comments Or Suggestions:**

N/A

**Other Strengths And Weaknesses:**

Strengths:

1. The paper is clearly written and easy to understand.

2. It provides theoretical and empirical evidence to justify that ACR is a poor metric for randomized smoothing.

Weaknesses:

1. The claim “the average certified radius (ACR) has emerged as the most important metric for comparing methods and tracking progress in the field” is not well substantiated. Please see my comments under “Claims And Evidence” and “Essential References Not Discussed.”

2. The theoretical claims lack originality. Please see my comments under “Theoretical Claims”.

**Questions For Authors:**

N/A

**Relation To Broader Scientific Literature:**

I am unaware of other works that study the weaknesses of evaluation metrics for certified robustness.

**Theoretical Claims:**

Proofs of theorems 1 and 2 are both correct. However, as mentioned earlier, these theorems are straightforward and the insights are unsurprising. Theorem 1 formalizes that a trivial classifier can achieve an arbitrarily high ACR by simply predicting the most likely class for all inputs, thereby having infinite certified radii for samples for this class and blowing up the ACR metric. Theorem 2 merely states that two points with the same L_2 norm have the same probability density under an isometric Gaussian distribution.

---

> ### Author Rebuttal · Authors · 2025-03-31
>
> We thank Reviewer $\Rg$ for the insightful review. We are happy that Reviewer $\Rg$ finds that our paper is easy to understand, and that our work provides both theoretical and empirical evidence to justify our conclusion. We address all concerns from Reviewer $\Rg$ below. We include new results, named with Figure S1 etc., in the [anonymized link](https://mega.nz/file/2NtyHIpA#EcvgiAMI7xMjXTgcGHnVpWdg-U2QnojAPVqF7peCMwM).
>
>
> **Q1: ACR is rarely used as a stand-alone metric. Is the claim that “ACR is the most important metric” proper?**
>
> We agree that claiming ACR is the most important metric is improper. To gain quantitative insights, we perform a literature survey in RS for publications on top conferences (ICML, ICLR, AAAI, NeurIPS), ranging from 2020 (since the first evaluation with ACR) to 2024. We only include works that use the certification algorithm proposed by Cohen et al. and propose general-purpose dataset-agnostic RS algorithms. We identify 13 works with the specified criteria, and report for each of them (1) whether ACR is evaluated, (2) whether universal improvement in certified accuracy is achieved at various radii, (3) whether a customized base model is used (either parameter tuning or architecture design) and (4) whether claims SOTA. The result is shown in Table S4.
>
> Among them we find 10 works that customize the model and claim SOTA, which is the focus of our study. 8 of them evaluate with ACR, and 7 of them claim SOTA solely based on ACR, i.e., claim SOTA without universal improvement in certified accuracy at various radii. Therefore, we conclude that ACR is of great importance to the field, and the practice of claiming SOTA based on ACR is widely spread. We will incorporate this study in the revised manuscript and revise the statement about the role of ACR accordingly.
>
>
> **Q2: The method proposed in this paper only has a marginal improvement on ACR compared to state-of-the-art (SOTA) approaches. Does this weaken the conclusion?**
>
> Our finding is that simply focusing on easy samples with the proposed Algorithm 2 replicate the progress in RS training strategies. Therefore, our method is **not** designed to achieve great advances in ACR further; instead, it is to prove how unreliable the evaluation of ACR could be. Therefore, we do not view the relatively small improvement over SOTA as a limitation of this work.
>
> **Q3: The theoretical analysis is straightforward and unsurprising. Does it mean that the paper lacks originality?**
>
> We agree that the theoretical analysis is straightforward but kindly disagree that it is unsurprising. First, to the best of our knowledge, this analysis has never been done before, leaving the weakness of ACR unknown yet. As a result, ACR is still widely adopted in RS training, as discussed in Q1. Second, the theoretical analysis serves as a crucial foundation for our empirical analysis and motivates our proposal to abandon ACR as the central metric for evaluating RS training algorithms. Therefore, we kindly disagree with the claim that our theoretical analysis lacks originality or significance.

---

### Official Review · Reviewer_nJsf · 2025-03-11

**Overall Recommendation:** 3

**Summary:**

This paper critiques the use of Average Certified Radius (ACR) as an evaluation metric for assessing the performance of certifiably robust classifiers, specifically focusing on randomized-smoothing-based approaches for robustness certification under the $\\ell_2$ perturbation threat model.

 (To give background: in an  $\\ell_2$-certifiably-robust classifier, for each input sample $x$, the classifier both returns a classification $\\hat{f}(x)$ as well as a radius $R(x)$, such that for any $x'$ such that  $\\|x-x'\\|_2 < R(x)$,  the sample $x'$ is guaranteed to be classified in the same way as $x$: that is, $\hat{f}(x) = \hat{f}(x')$. Typical randomized smoothing approaches (for the $\\ell_2$ metric) compute $\hat{f}(x)$ by taking the plurality-vote of the output of a _base classifier_ $f(x + \delta)$ on many noise instances $\delta$ drawn from an isometric Gaussian distribution: the certified radius $R(x)$ is then computed as a (monotonically increasing) function of $p_A$, the fraction of noise instances $\delta$ on which $f(x + \delta)$ retuned the plurality class.)


As an evaluation metric, the Average Certified Radius is defined on a labelled  test set $(x_1, y_1), ..., (x_m,y_m)$ as:

$$
ACR = \\frac{1}{m} \\sum_{i=1}^m \\begin{cases} R(x_i) & \\text{ if } f(x_i) = y_i, \\\\ 0 & \\text{ otherwise. }\\end{cases}
$$

The paper claims that ACR is commonly used to evaluate randomized-smoothing based certifiably robust classifiers. The paper then notes (Theorem 1)  that ACR can be ``gamed'' to be made arbitrarily large: specifically, a _constant_ function $\hat{f}(x) = c$, which classifies all samples with the same class c, combined with a certification technique that (_correctly_) reports an arbitrarily large certified radius R(x) for each sample (note that these certifications are correct, because the classification function $\hat{f}(x)$ is constant) will have an arbitrarily large ACR as long as *any* samples  in the test set are labeled as c. In particular, this set-up can be instantiated using standard randomized smoothing certification techniques, by using a constant base classifier, and a sufficiently-large number of smoothing samples to make R(x) as large as desired.

In practice, in randomized smoothing (RS) certification techniques, the certified radius R(x) is *capped* by a function of the number of smoothing samples $\delta$ used, and hence by the certificate computation time. However, the paper goes on to claim that in RS techniques, training design choices that increase ACR tend to increase the certified radii R(x) of the "easiest" samples in the data distribution (those which already have a large certified radius) at the expense of "harder" samples. This phenomenon is demonstrated empirically by comparing the cumulative certified radius distribution of Cohen et al. (2019), one of the first RS techniques, with the cumulative  certified radius distributions from more-recent works, on CIFAR-10 (Figure 6; Table 2). For small radii r, Cohen et al's method is more likely to correctly classify samples and certify them as robust to radius R(x) >= r, while for large radius r, the more recent techniques prevail. The paper explains this phenomenon theoretically as being due to the fact that the certified radius R(x) grows very quickly as a function of  $p_A$ when $p_A$ is near 1 ( and $R(x)$  is already  large ), while growing much more slowly with  $p_A$ when $p_A$ and $R(x)$ are small. Therefore increasing the certified radii of "easy" samples even further is "easier" than increasing the certified radii of "hard" samples by the same amount, because it requires increasing $p_A$ of the "easier" samples by a much smaller increment.


The paper goes on to design a training method for certifiably robust classifiers that is designed to "game" ACR as much as possible (practically; without increasing the number of smoothing samples).  They propose to skip "hard" samples during training, weight training samples by "easiness," and adversarially choose smoothing perturbation vectors $\\delta$ during training to ensure that $p_A$ is as close to 1 as possibly for "easy" samples. The resulting classifiers have state-of-the-art ACR on CIFAR-10 and relatively-low clean accuracy (although not the lowest among certifiably-robust methods.

Finally, the paper suggests alternatives to using ACR to evaluate certifiable robustness: reporting the cumulative distribution of R(x) (that is, the fraction of test samples which are correctly classified and have $R(x) \geq r$ for various values of $R$) as well as the distribution of $p_A$.

**Claims And Evidence:**

The basic argument that ACR alone provides an incomplete picture of certifiable robustness, because it can be made arbitrarily high by using a trivial (constant) classifier, is correct, and the proof of the associated Theorem 1 is correct.

However, the submission claims that:

"Randomized smoothing is a popular approach for providing certified robustness guarantees against adversarial attacks, and has become an active area of research. Over the past years, the average certified radius (ACR) has emerged as the **most important** metric for comparing methods and tracking progress in the field." [Abstract] (emphasis added)

"Average Certified Radius (ACR), defined to be the average of the certified radiuses over each sample in the dataset, has been **the main metric** to evaluate the effectiveness of these methods" [Introduction, Lines 31-33] (emphasis added).

These claims are not well-supported. While the paper does cite six works which report ACR as a metric, the following additional context is relevant and is not mentioned:
- **None** of the works report ACR _alone_ as the sole metric for comparing RS techniques; all of them also give the cumulative distribution of R(x).
- All but two of the cited works which use ACR share the same first-author, showing that the use of this statistic is less wide-spread than is suggested by the total number of provided references.
- Several works which deal with $\\ell_2$ certifiable robustness via randomized smoothing, which do not use ACR, were not mentioned. This includes works which propose new techniques, such as:

Awasthi et al. "Adversarial robustness via robust low rank representations", NeurIPS 2020

Zhang et al. "Black-Box Certification with Randomized Smoothing: A Functional Optimization Based Framework" NeurIPS 2020

Li et al. "Double Sampling Randomized Smoothing" ICML 2022

as well as survey papers which compare many proposed techniques, and do **not** report ACR, such as:

Li et al. "SoK: Certified Robustness for Deep Neural Networks" IEEE SP 2023

Furthermore, the claims in the Abstract and Introduction sections highlighted above are not qualified as being only about $\\ell_2$ certifiable robustness, although later it is stated that " In this work, we focus on the L2 neighborhood of an input," (Background, lines 66-67). The cited examples of ACR are all papers about L2 certified robustness, and no examples are given outside of this specific line of work. However, many papers exist which use randomized smoothing-based techniques and do not report ACR.  (See the above survey for examples of certification results for other metrics.) Therefore the general claim made in the abstract that "Randomized smoothing is a popular approach [...] the average certified radius (ACR) has emerged as the **most important** metric for comparing methods and tracking progress in the field"  is not supported by evidence provided in the paper. To properly evidence this claim, one would need to perform a much wider and more objective meta-analysis of all literature in randomized smoothing. (Note that even if *every* paper using randomized smoothing reported ACR, this would not justify the claim that ACR is the *most important* metric for comparing methods.)

Additionally, while the experimental evidence does seem to justify that, for fixed smoothing noise standard-deviation $\sigma$, newer RS methods tend to increase the certified robustness of "easy" samples at the expense of accuracy on "hard" samples (at least on CIFAR-10), the experimental comparison includes SmoothAdv (Salman et al. 2019), which does not report ACR, but which seems to behave similarly in terms of the shape of the certificate distribution to the methods which do, at least compared to Cohen et al (2019). This seems to undermine the **causative** claim in the abstract that "Overall, our results suggest that ACR has introduced a strong undesired bias to the field."

The other claim that this paper makes that I do not believe is sufficiently justified is the relevance of the theoretical argument made in Section 4.2; specifically, the claim that "We have shown that ACR strongly prefers easy samples in §4.2." (line 155-156) The issues with this claim are elaborated below under "Theoretical Claims."

**Essential References Not Discussed:**

See the above discussion. While, to my knowledge, this is the first work to explicitly identify that ACR is a problematic metric, there are many works in this space that do not use ACR but were not cited. To restate some examples given above:

Awasthi et al. "Adversarial robustness via robust low rank representations", NeurIPS 2020

Zhang et al. "Black-Box Certification with Randomized Smoothing: A Functional Optimization Based Framework" NeurIPS 2020

Li et al. "Double Sampling Randomized Smoothing" ICML 2022

Li et al. "SoK: Certified Robustness for Deep Neural Networks" IEEE SP 2023

There are many more such works in existence; a more comprehensive survey is needed to assess whether ACR is truly a ubiquitous metric.

**Experimental Designs Or Analyses:**

One confusion I had with the experimental design was the explanation of Figure 3 in line 193: " As a result, Gaussian training has higher P(pA ≥0.5) (clean accuracy)," Clean accuracy is not the same as p_A >= 0.5. If there are more than two classes, samples can be labeled correctly  by the smoothed classifier with p_A < 0.5; they just won't get a certified radius by Cohen et al's method. It would be good to also label the _actual clean classification accuracy_ of each classifier. (For example, if the classifiers are classifying samples correctly, with radius 0)

Additionally, throughout the paper, it is implicitly assumed that we are using the empirical certification algorithm described in Cohen et al, which only bounds the probability $f(x+\delta)$ returning the ``top'' class. For example, on Line 137-138: " Further, when pA < 0.5, the data point will not contribute to ACR at all," Some methods can in fact certify when p_A < 0.5, such as the method used in (Lecuyer et al, 2019; Dvijotham et al., "A Framework for robustness Certification of Smoothed Classifiers using F-Divergences" ICLR  2020). This should be clarified.

**Methods And Evaluation Criteria:**

The methods and evaluation appear to be sound; however, only CIFAR-10 was used in the experiments. It would be better to include a dataset with more natural images (for example ImageNet, or, if that is not feasible, an ImageNet subset).

**Other Comments Or Suggestions:**

Minor Comments:


Line 34-35: "Although some studies refrained from using ACR in their evaluation unintentionally" -- It is not knowable whether on not this was "intentional"; it would be better to say "Although some studies have not used ACR (without mentioning a justification for this choice)"

Line 44: "Due to the incompleteness of adversarial attacks ": I think this means that adversarial attacks only prove an upper bound on the true robustness/distance to the decision boundary, but this is not worded clearly.

Line 31" "radiuses" -> radii

Line 96: "is commonly used" -> "was commonly used"

Equation on line 94: The sum is meaningless here, and so can be omitted: we're taking the average over m identically-distributed samples, within an expectation. The sum can be commuted with the expectation, where it then becomes clear that all of the summands are equal.

Line 106: "While they all improve ACR" -> "While these methods all improve ACR"

Line 134-135: "with minimal robustness on at least half of classes." Why not minimal robustness on all but one class?

Lines 138-139: "since inputs with lower pA require a larger budget to certify. " : This seems to be an oversimplification. On the contrary, isn't the certified radius most sensitive to the estimated probability, and hence to sampling budget, when pA is near 1? (The example given in this paragraph shows a more complex relationship)

Line 160: "We follow the standard certification setting in the literature, setting N = 10^5 and α = 0.001." : I believe these parameters are originally from Cohen et. al. 2019; I would cite them.

Section 4.2 (and Figure 2): I believe this figure is showing the relationship between the _measured_ p_A (from samples) and the certified radius. However, p_A is actually defined as the _population average_ in line 79 in Section 3. This should be clarified.


Table 1: is this _parameter_ gradient or _feature_ gradient magnitude?

Line 319-321: "Therefore, with the adaptive attack, Gaussian training obtains a similar gradient norm distribution to SOTA algorithms": I would no longer call the proposed method, with the adaptive attack, "Gaussian training"

"Furthermore, the community should also consider randomized smoothing methods with σ as a variable rather than a hyperparameter, thus effectively removing the dependence on σ. While there are some preliminary works in this direction (Nurlanov et al., 2023), they usually break the theoretical rigor, thus more research is needed in this direction." Nurlanov et al is not a relevent citation (it does not use randomized smoothing.) There are works in this direction, however, as noted, there are issues with correctness in many of these works. see (Sukenik et al,"Intriguing Properties of Input-Dependent Randomized Smoothing" ICML 2022) for further discussion.


Use of capital I for both identity matrix and indicator function is confusing; would use bold 1 instead for indicator function.

Proof of Theorem 1: the proof seems to implicitly be using binomial (Clopper-Pearson) confidence interval; I would be more explicit about it.

**Other Strengths And Weaknesses:**

Strengths:
- The main argument of the paper, that ACR is not an appropriate metric on its own, because it may be vacuous, is compelling and is argued correctly.
- While this was not the stated aim of this paper, it appears from the results that the proposed training method (for fixed $\sigma$) did achieve a higher certified accuracy at large radii than all prior works, at least on CIFAR-10. This training method can then be seen on its own as an additional contribution.

**Questions For Authors:**

- Is there any _quantitative evidence_ that the _proportion of published randomized smoothing papers which use ACR_ is above 50\%? If so, how many use ACR alone?

- Does the observed trend apply to datasets other than CIFAR-10?

++++++++++++++++++++++
Both questions were responded to during rebuttal period: I am increasing my score to 'Weak accept'.

**Relation To Broader Scientific Literature:**

This paper provides concrete suggestions to future researchers in randomized smoothing for certified adversarial robustness, specifically:
1. To avoid using ACR (Average Certified Radius) as a metric for robustness.
2. To instead report the highest-achievable certified accuracy at specified radii.
3. To report the cumulative distribution of p_A for the test set.

For (1) the paper does establish that at least some prior works do report ACR, and then shows how this statistic can be vacuous; furthermore, I am not aware of any prior works that explicitly point out these issues with ACR. However, it is not well-established that the use of ACR is as common as claimed ("the most important metric") . Additionally, as noted above, (2) is already done _nearly-universally_ in prior works.  Furthermore, (3) may be somewhat limited: if methods differ in terms of _how_ they compute certificates given $p_A$, then (3) does not allow for a fair comparison (for example: Li et al. "Double Sampling Randomized Smoothing" ICML 2022); additionally, (3) does not allow for comparison to non-RS certified robustness methods. Furthermore, p_A is not actually relevant to downstream applications: it is an internal metric of the certification process.

**Theoretical Claims:**

The main theoretical claim in the paper (that ACR can be unbounded for trivial classifiers), Theorem 1, is correct and has a correct proof.

Theorem 2 is a minor technical result, which is not of great importance to the paper. However, its proof is only correct as written if we interpret $P_{\\mathcal{N}(0,\\sigma^2I_d)}(\\delta)$ to refer to the probability _density_ at $\delta$. This notation is not explained, and conflicts with the notation in the Background section, where  $P_{\\delta \\sim \\mathcal{N}(0,\\sigma^2I_d)}(f(x+\\delta) = c)$ refers to the _probability_ (not the density) of the event. The text at line 305-306: "This is because for every δ∗such that ∥δ∗∥2 = ∥δ0∥2, the probability of sampling δ∗is the same as δ0. We formalize this fact in Theorem 2." should also be revised: "the probability of sampling δ∗" is _infinitessimal_.

A major theoretical over-claim in this work is that:  "We have shown that ACR strongly prefers easy samples in §4.2." (line 155-156). The theoretical argument in Section 4.2 shows that the certified radius R(x) grows very quickly as a function of $p_A$ when $p_A$ is near 1 ( and $R(x)$  is already large ), while growing much more slowly with  $p_A$ when $p_A$ and $R(x)$ are small. Therefore, it is suggested that algorithms which are tuned to increase average radius will tend to increase p_A for samples where p_A is already large, because this will have a greater impact on average radius than an equal increase in p_A for samples where p_A is small. However, this theoretical observation alone does *not* establish, as implied, that it is "easier" to increase the certified radius of a sample with p_A near 1 than a sample with a lower p_A. Concretely, while it is true that increasing p_A from .99 to .999 increases R(x) by a much larger margin than increasing  p_A from .6 to .609, it is not obvious that it is _as easy to increase_ p_A from .99 to .999 as to increase  p_A from .6 to .609. In fact, one can make a compelling theoretical argument for the opposite: that it should be much harder to increase p_A from .99 to .999 than to increase p_A from .6 to .609.

Specifically, due to the nature of the high-dimensional Gaussian distribution, the minimum _volume_ around $x$ for which the base classifier $f(x+\\delta)$ must return class A in order to achieve $p_A = .999$ is much larger than the minimum volume around $x$ that must belong to class A in order to achieve $p_A = .99$, and in particular,  this gap in minimum-volumes is much larger than the gap between the volumes required to achieve $p_A = .6$ and $p_A = .609$. (To help understand this, note that achieving p_A = 1 _requires_ that $f$ is constant _everywhere_). Therefore an analysis in terms of $p_A$ alone is perhaps misleading, and does not fully "show that ACR strongly prefers easy samples".

The empirical evidence _does_ seem to suggest that the methods with the highest ACR tend to focus on increasing p_A for "easier" samples; however, I think it is an over-claim to say that the argument in Section 4.2 fully explains the phenomenon: more nuance is required.

---

> ### Author Rebuttal · Authors · 2025-03-31
>
> We thank Reviewer $\Rn$ for the insightful review. We are happy that Reviewer $\Rn$ finds our work is important and valuable, and points out imperfect expressions. We will address all the concerns raised by Reviewer $\Rn$ in the following. We include new results, named with Figure S1 etc., in the [anonymized link](https://mega.nz/file/2NtyHIpA#EcvgiAMI7xMjXTgcGHnVpWdg-U2QnojAPVqF7peCMwM).
>
>
> **Q1: Is ACR the most important metric in the community? Is there any quantitative evidence which supports their claim?**
>
> Please refer to our reply to Q1 of Reviewer $\Rg$.
>
> **Q2: SmoothAdv, which is before the invention of ACR, has similar issues as presented in this paper. Does this undermine the claim that ACR introduced a strong undesired bias to the field?**
>
> We agree that although the empirical evidence in Sec 4.3 shows that a strong bias is introduced to the field, it does not directly imply that ACR is the only reason. In fact, the hypothesis that ACR has causality with the bias is not provable. Therefore, we will temper our tone and state that “A strong bias has been introduced to the field, which is highly likely due to the practice of claiming SOTA based on ACR given the demonstrated properties”.
>
> **Q3: Could the authors clarify notations in Theorem 2 and the text at 305-306 rc?**
>
> We thank Reviewer $\Rn$ for pointing out the notation conflicts. We will define the notations more clearly and replace the infinitesimal probability with PDFs of the Gaussian function in the revised manuscript.
>
> **Q4: Easier samples contributing much more to ACR than harder ones does not imply that it is easier to improve $p_A$ of easy samples than hard ones. Is the claim that ACR strongly prefers easy samples proper?**
>
> We agree that more contribution from easy samples does not directly imply that focusing on easy samples will increase ACR. We refer to our reply to Q3 of Reviewer $\Ru$ for a detailed discussion. However, the claim that ``ACR strongly prefers easy samples`` simply states that easy samples contribute more, but does not conclude that focusing on easy samples will increase ACR. Nevertheless, to avoid confusion, we will adjust this statement properly.
>
> **Q5: Do the results generalize to other datasets?**
>
> Please refer to our reply to Q1 of Reviewer $\Rp$.
>
> **Q6: Does this work assume applying the certification algorithm described in [1]?**
>
> Yes. In fact, we only discuss works that are based on the certification algorithm in [1] since the subject is RS training strategies but not certification algorithms. We will clarify this in the revised manuscript.
>
> **Q7: Could the authors clarify the definition of clean accuracy when not using the prediction algorithm of [1]?**
>
> We consider the certification method in [1], which does not provide a prediction if the empirical $p_A$ is below 0.5 (i.e., the algorithm returns ABSTAIN). In this case, the clean accuracy is defined as the accuracy at a certified radius of 0. This convention is also adopted in previous works such as [1, 2, 3].
>
> As a future reference, we additionally report the accuracy of different methods when the model performs a majority vote without abstaining in Table S3. It shows that without abstaining the clean accuracy is always marginally higher than that with abstaining. The result without abstaining is consistent with the analysis in this work performed with abstaining.
>
> **Q8: For the proposed metric, the cumulative distribution of $p_A$, can it be used to compare different certification algorithms? How to use it for downstream applications?**
>
> As discussed in Q6, this work focuses on the certification algorithm proposed by [1]. However, cross-certification evaluation is still possible. Appendix B thoroughly discusses how to convert the distribution of $p_A$ into certified accuracy at different radii and different budgets. If comparisons of different certification algorithms are desired, then based on methods discussed in appendix B.1, one may first convert it to certified accuracy before comparison. This is more flexible than certified accuracy at fixed budget, because different certification might vary in certification budget. Similarly, for downstream applications, since the proposed metric is more generalized than certified accuracy, it allows more flexible evaluation; importantly, every evaluation that is based on certified accuracy can be recovered cheaply based on the distribution of $p_A$. Nevertheless, this metric is only meaningful for $p_A$-based certification; it must be converted into other metrics to compare with other certification algorithms. We will clarify these aspects in the revised manuscript.
>
> Reference
>
> [1] arxiv.org/abs/1902.02918
>
> [2] arxiv.org/abs/1906.04584
>
> [3] arxiv.org/abs/2212.09000

---

> > ### Comment · Reviewer_nJsf · 2025-04-07
> >
> > Thank you for your response. The additional, systematic literature review is helpful and the results on ImageNet are also appreciated. I am raising my score on the strength of these additional results. (I also expect that the over-claim about ACR being the "most important" metric will be revised in the final draft, as promised; and for the final draft to more clearly state that the work is _only_ concerned with training methods for Cohen et al.'s certification scheme, not Randomized Smoothing in general.)
> >
> > For Q2 above, I would directly call out the fact that SmoothAdv does not use ACR, but shows the same trend, in the paper.
> >
> > I don't think that the response to Q4 really answered my concerns. It is not clear what "easy samples contribute more" in the response means: ACR is an unweighted average, so each sample contributes the same. The argument in Section 4.2 concerns the _marginal_ effect of a small _increase_ of $p_A$ on ACR. I pointed out that this argument may have little relevance to ACR, because increasing $p_A$ on  any "easy" sample by a given $\epsilon$ may be much more difficult than increasing $p_A$ on  a "hard" sample by the same $\epsilon$. In fact, we might expect this to be the case, because the additional volume of $f$ that must be near-constant around $x$ to increase $p_A$ from $0.99$ to $0.99 +\epsilon$ is much greater than the additional volume of $f$ that must be near-constant around $x$ to increase $p_A$ from $0.6$ to $0.6 +\epsilon$.  Therefore a theoretical analysis in terms of $p_A$ alone is incomplete. (The response to uKgY which was cited does not mention Section 4.2 at all, which is what my question was about.)
> >
> > Regarding  Question 7: Cohen et al propose _two_ algorithms: PREDICT and CERTIFY. PREDICT returns the classification of a sample, while CERTIFY gives a certified radius. Both algorithms can abstain. However, while CERTIFY will always abstain of $\hat{p}_A < 0.5$, PREDICT does not necessarily abstain when there is no majority class: as long as as there is a sufficient gap between the top class and the runner-up class, PREDICT will return the top class (the winner of the plurality vote.) Based on this scheme, the "Clean Accuracy" should refer to the accuracy of PREDICT, which is distinct from either the fraction of samples for which  $\hat{p}_A \geq 0.5$ *or* the "no abstain" top class.

---

> > > ### Author Response · Authors · 2025-04-08
> > >
> > > We thank Reviewer $\Rn$ for the insightful reply and for appreciating our additional literature study and experiments. We are happy to provide further discussion on Q4 and Q7.
> > >
> > > **Further discussion on Q4**
> > >
> > > It seems that Reviewer $\Rn$ refers to different aspects from ours. We believe that Reviewer $\Rn$ refer to the fact that the derivative of ACR to each radius is the same ($1/n$), but we refer to the fact that the derivative of ACR to $p_A$ is dramatically large when $p_A$ is close to 1. We may further improve the writing clarity regarding this aspect.
> > >
> > > **Further discussion on Q7**
> > >
> > > We may further clarify the aspects about clean accuracy since different readers may have different interpretations about this in the context of RS and Cohen’s prediction algorithm.
> > >
> > > We thank Reviewer $\Rn$ for their efforts in evaluating our work. We hope our reply has fully addressed their concerns.

---

### Official Review · Reviewer_Ptgp · 2025-03-11

**Overall Recommendation:** 4

**Summary:**

The authors make a strong claim that the Average Certified Radius (ACR) - which is widely used through the Randomized Smoothing (RS) community - is not a good metric at all for a number reasons. They prove it, and provide the ways how it can be exploited for improving ACR.

## update after rebuttal
Authors provided the asked additional experiments (even for ImageNet), so I keep my original score.

**Claims And Evidence:**

Claims:
* Theoretic proof that ACR can be arbitrarily large even for a trivial classifier, esp. when its improvement is rooted in easy examples
* Empirical justification of the current RS training strategies intention to concentrate on easy examples (with high $p_{A}$)
* Based on theoretical and empirical observations, the authors proposed the specific methods to improve the existing RS training strategies (Section 5) by a) discarding hard examples during training, b) training examples re-weighthing with Certified Radius, and c) adjusting the perturbation to have the same norm but broking the classifier (they called it "Adaptive Attack")

Overall, the approach by authors can be formulated as following:
1. They empirically and theoretically investigated on why the ACR is a bad metric
2. They exploited their observations to produce the best (SotA) defense methods
3. They proposed what could be the better metric instead of ACR (Section 7 - e.g., using the best certified accuracy at various radii or use $\sigma$ as a variable, not a hyperparameter)

**Essential References Not Discussed:**

NA

**Experimental Designs Or Analyses:**

Actually, two main remarks:
1. No usage of other than CIFAR-10 dataset. ImageNet was used even in the seminal paper of [1].
2. There is no re-estimation of the existing RS defenses for the metrics proposed in Section 7. Yes, there is a slight approach in the Appendix B, but it is definitely not enough. If the paper proposes the new way of measuring Certified Robustness for RS, than it makes to provide the new measurements of the methods and see how they compare against each other - whether the order is different etc

[1] Cohen, J. M., Rosenfeld, E., and Kolter, J. Z. Certified adversarial robustness via randomized smoothing. In Proc. of ICML, volume 97, 2019.

**Methods And Evaluation Criteria:**

All the proposed methods (already existing and adopted in the community of RS researchers as well as the new one proposed in Appendix B for constructing the Empirical CDF for $p_{A}$) sound solid and reasonable.

As for the datasets itself, CIFAR-10 was used - which is quite widely used for the task of RS. Unfortunately, no ImageNet results were delivered - which would provide a stronger message about this paper.

**Other Comments Or Suggestions:**

I'm not quite sure if it is a typo or some my misunderstanding of the following sentence in Section 7:

"this represents the setting where one first fixes an interested radius, and then try to develop the best model with the highest certified radius at the pre-defined radius."

**Other Strengths And Weaknesses:**

To me, very well organized paper with the structure: Observation --> Exploitation --> Proposal to change.

Would be nice to address the items mentioned in "Experimental Designs Or Analyses"

**Questions For Authors:**

NA

**Relation To Broader Scientific Literature:**

The whole paper is devoted to the certified robustness through probabilistic approach - randomized smoothing. RS is the main method to assess certification of deep NNs.

**Theoretical Claims:**

Yes, the authors provided the proof of the Theorem 1 (about arbitrary high ACR) in Appendix A1, as well as the auxiliary simple proof for the Theorem 2 (about the probabilities of the equal norm perturbations). While looking simply the theorems are carefully proved.

Moreover, inside the Appendix B.1 there is a description (while not explicitly theoretically formulated but still rigorous) of the procedure to convert ECDF($p_{A}$) to ECDF($r_{cert}$) with different value of $N$ which is very well described.

---

> ### Author Rebuttal · Authors · 2025-03-31
>
> We thank Reviewer $\Rp$ for the insightful review and faithful interpretation of our work. We are happy that Reviewer $\Rp$ finds our work important, sound and solid. We will resolve all the concerns below. We include new results, named with Figure S1 etc., in the [anonymized link](https://mega.nz/file/2NtyHIpA#EcvgiAMI7xMjXTgcGHnVpWdg-U2QnojAPVqF7peCMwM).
>
> **Q1: Do the results generalize to ImageNet?**
>
> Yes. We perform a new analysis on ImageNet, generalizing the findings presented in this work. Concretely, Figure S1 is an extension of Figure 3 on ImageNet, and Table S1 is an extension of Table 2 on ImageNet. The hyperparameters used in the study are reported in Table S2. We find that all our conclusions remain correct on ImageNet, justifying their generalization across datasets.
>
> **Q2: Could the authors extend the evaluation of previous works using the proposed “ECDF of $p_A$” metric?**
>
> Sure. The ECDF of $p_A$ on CIFAR-10 with different $\sigma$ and algorithms are shown in Figure S2. We note that an algorithm is better only if it has higher ECDF for all $p_A \ge 0.5$. Therefore, it is possible that neither of two algorithms is better than the other.
>
> For various $\sigma$, Gaussian is always the best in the low $p_A$ region, i.e. from $p_A=0.5$ to around 0.65. Although CAT-RS consistently has the highest ACR under all $\sigma$, it only outperforms other methods in the high $p_A$ region. For example, with $\sigma=0.5$, CAT-RS is the second worst method when $p_A$ is below 0.74. We list some cases where two methods have strict orders: with $\sigma=0.25$, CAT-RS is better than Consistency and SmoothAdv; with $\sigma=0.5$, CAT-RS is better than SmoothMix; with $\sigma=1.0$, CAT-RS is better than SmoothMix and MACER.
>
> **Q3: Clarification of the typo in Sec. 7.**
>
> Thanks for pointing out the typo. It should be "try to develop the best model with the highest **certified accuracy** at the pre-defined radius".

---

> > ### Comment · Reviewer_Ptgp · 2025-04-02
> >
> > I would like to thank authors for making additional experiments and addressing my remarks about experimental part.
> >
> > While the curves look similar, I'm a little bit concerned about the Table 2 in the anonymized link. It seems that a list of your "exploitation" techniques like "Discard Hard Data During Training", "Data Reweighting with Certified Radius", and "Adaptive Attack on the Sphere" is mostly overfitted on CIFAR-10 - because for ImageNet, the ACR (that should be the best for your approach) is now even not the second biggest one for high $\sigma$. It makes the sections of "Replicating the Progress in ACR" in the original paper look questionable.
> >
> > Looking forward to any insights about it.

---

> > > ### Author Response · Authors · 2025-04-04
> > >
> > > We thank Reviewer $\Rp$ for the quick reply and their effort in reading our new experimental analysis; their comments have been constructive and encouraging. We are glad to provide further discussion about Table S1 below.
> > >
> > > The relevant claim made in this manuscript is that simply focusing on easy samples is enough to replicate the advances in RS training strategies. We agree that while on CIFAR-10 the proposed modification to Gaussian training achieves SOTA ACR universally, on ImageNet it only replicates a large portion of the ACR advances. Concretely, for $\sigma=0.25$, the proposed method recovers (0.529 - 0.476) / (0.532 - 0.476) = 94.6% of the advance; for $\sigma=0.5$, it recovers (0.842 - 0.733) / (0.846 - 0.733) = 96.5%; for $\sigma=1$, it recovers (1.042 - 0.875) / (1.071 - 0.875) = 85.2%. Therefore, it is confirmed that the same evidence is found on ImageNet. To be more precise, we will change the claim to “simply focusing on easy samples is enough to replicate the advances in RS training strategies, sometimes even surpassing existing SOTA algorithms”.
> > >
> > > Further, there are complex reasons why our preliminary results on ImageNet provided in the initial rebuttal do not surpass SOTA, and more efforts may further improve it. First, on ImageNet CAT-RS utilizes a pre-trained model [1] to determine the inclusion of certain loss terms, which is not used on CIFAR-10. This improves the result of CAT-RS on ImageNet, while our preliminary results even skipped “Data Reweighting with Certified Radius” to deliver fast results.  Second, we did not conduct a sufficient hyperparameter search on ImageNet due to time reasons in the initial rebuttal, while a good tuning is performed on CIFAR-10. Despite these challenges, our proposed method still recovers most of the advances on ImageNet. Therefore, we believe that this proves the generalization of our results across the dataset. In the final manuscript, we will apply more computation on ImageNet and check if similar tricks using a pre-trained model may be applied for our method, e.g., use the pre-trained model to determine the hardness of inputs throughout training rather than using $p_A$ computed on-the-fly. This should further improve our preliminary numbers, with the potential to also establish SOTA ACR on ImageNet. We also would like to note that due to the arguments made in this manuscript, achieving SOTA ACR is less meaningful, and our current results are sufficient to support our claims.
> > >
> > > Reference
> > >
> > > [1] arxiv.org/abs/2212.09000

---

### Official Review · Reviewer_ukgY · 2025-03-13

**Overall Recommendation:** 3

**Summary:**

The authors investigate the validity of the Average Certified Radius (ACR) as a measure for robustness.

**Claims And Evidence:**

Claims and Evidence:

C1. Authors theoretically  show that with a large enough certification budget, ACR of a trivial classifier can be arbitrarily large, and that with the certification budget commonly used in practice, an improvement on easy inputs contributes much more to ACR than on hard inputs, more than 1000x in the extreme case.

E1. One of the evidence for this claim provided by the authors is Theorem 1.  I am not convinced of this result being meaningful. Indeed a constant classifier is trivially the most robust classifier by any measure of robustness. And Theorem 1 formalizes this intuition.  But this type of analysis is fundamentally flawed in the methodology: simultaneously optimizing two objectives is not the same as optimizing both of them separately. That is optimizing only robustness will not lead to the same solution as the one when optimizing for robustness and accuracy together.This analysis does not take into account predictive performance of the classifier, and any real-world robustness training algorithm will optimize for both robustness and accuracy.

## Update after rebuttal: The authors correctly pointed out my misjudgment of the theorems results, and I think there analysis is correct and meaningful.

C2.  ACR disproportionately focus on easier examples, i.e., samples with more confident predictions (and hence larger p_A) are disproportionately represented in ACR. However, the authors also claim that this leads to potentially poorer RS algorithms (which I do not agree with, see below).

E2. As ACR is essentially an average, where the weights are the proportional to CDF of a normal distribution at $p_A$, it indeed over-represents samples with larger $p_A$. However, in section 4.3, I do not agree with the analysis. The authors criticize the fact that the RS training procedures "lowers the $p_A$ for harder samples" --- I do not understand why they should behave otherwise? I would argue that the user should see a lower $p_A$ for harder samples, this does not contradict the fact that ACR can be misleading, but claiming that this has lead to wrong RS algorithms is misleading.

## Update after rebuttal: I think the situation is clearer to me now. And I think the point about "selection bias" seems valid.

C3. The authors show that focusing only on easier samples can improve ACR.

E3. Authors provide an algorithm and experimental evaluation to this end. However, I think this fact is clear, as ACR is a weighted average as mentioned above. See "Experimental Designs Or Analyses".

## Update after rebuttal: My statement was wrong, and the authors are indeed correct to provide experimental evidence for gaming ACR.

**Essential References Not Discussed:**

None, to the best of my knowledge.

**Experimental Designs Or Analyses:**

I think the experimental design is sound, but it is also tautological, i.e., the hypothesis they are testing is not falsifiable by the experiments they are performing. In my understanding, given the definition of ACR, and then discarding hard samples one will provably improve ACR.


## Update after rebuttal: I think this assessment was indeed incorrect and my understanding of the paper has been improved by looking at section 4.2 again, and reading the author's discussion with Reviewer nJsf.

**Methods And Evaluation Criteria:**

Authors provide theoretical results, and verify them experimentally, with an algorithm to produce the misleading ACR behavior, on widely known benchmark datasets and RS.

**Other Comments Or Suggestions:**

-- I am quite ambivalent about the paper, on one hand I think the key message of the paper is pertinent. However, as described above, I find many points raised in the paper to be rather misleading (in my understanding).

## Update after the rebuttal: As mentioned earlier, I do not think anymore that the analysis is misleading, barring the fact that ACR still does not seem to be the most important metric

**Other Strengths And Weaknesses:**

Strengths:
The key message of the  paper is potentially very important.


Weakness:
Almost all of the peripheral analysis is potentially misleading.
The main message should just be that average is not a robust statistic and one should not base metrics for safety-critical applications on that.

## Update after rebuttal: I would like to retract my comments about the "misleading analysis". However, I still believe that the main result indeed, in large parts, can be attributed to averages not being robust statistics.

**Questions For Authors:**

-- Why cant the entire message of the paper be summarized in the fact that ACR is average of the radii, and averages can be arbitrarily moved by just moving one instance, and hence ACR can be made arbitrarily good by just focusing on easy samples with large robustness radii?

-- What is the usefulness of Algorithm 2? as one can see from definition of ACR focusing too much on easy samples will indeed improve it. Is Algorithm 2 verifying this point?

**Relation To Broader Scientific Literature:**

The main message of the paper is potentially very relevant to RS research, depending on how widely ACR is used as a metric.
However, I think ACR is a misleading metric only if it is used as a stand-alone metric.

The fundamental point of the paper can be boiled down to the fact that average, and hence ACR, is not a robust statistic.

## Update after rebuttal: I think this assessment is indeed correct to a large extent.

**Theoretical Claims:**

See above in "Claims and Evidence" section

---

> ### Author Rebuttal · Authors · 2025-03-31
>
> $\newcommand{\Ru}{\textcolor{green}{uKgY}}$
> $\newcommand{\Rp}{\textcolor{blue}{Ptgp}}$
> $\newcommand{\Rn}{\textcolor{fuchsia}{nJsf}}$
> $\newcommand{\Rg}{\textcolor{purple}{gJ9M}}$
>
> We thank Reviewer $\Ru$ for the insightful review. We are happy that Reviewer $\Ru$ finds our work important and experiments sound. In the following, we will address all concerns raised by Reviewer $\Ru$. We include new results, named with Figure S1 etc., in the [anonymized link](https://mega.nz/file/2NtyHIpA#EcvgiAMI7xMjXTgcGHnVpWdg-U2QnojAPVqF7peCMwM).
>
>
> **Q1: Does the robustness in this paper (especially Thm 1) consider both predictive performance and output stability under small input perturbation?**
>
> Yes. As defined in 68-69 rc, the robustness defined in this work considers both the predictive performance and the output stability. Specifically, Thm 1 also considers both aspects, and inputs that are mistakenly predicted will not contribute to ACR. Furthermore, Reviewer $\Ru$ claims that ``a constant classifier is the most robust classifier by any measure of robustness``; this is not true. For example, under our definition of robustness, a constant classifier can have zero robustness: consider a dataset that only contains class 0 for a binary classification task, then a constant classifier predicting class 1 will have zero robustness (and zero ACR). We hope this classifies the misconception about our definition of robustness.
>
> **Q2: Why shouldn’t RS training lower $p_A$ for harder samples?**
>
> The term ``harder samples`` in Sec 4.3 refers to samples with $p_A$ larger than but close to 0.5, and ``easier samples`` refers to samples with $p_A$ close to 1. Therefore, as shown in Sec 4.3, if RS training is justified to lower $p_A$ for harder samples, then under small radii (e.g., $r=0$), the certified accuracy of “better algorithms” will be **lower** than “worse algorithms”, contrary to the intuition.
>
> In addition, the claim made in Sec 4.3 is not “this has led to wrong RS algorithms”; instead, we claim that this introduces selection bias (that “better algorithms” have worse certified accuracy under small radii) into the development of RS algorithms, as explicitly stated in the title of Sec 4.3. The latter is directly supported by empirical evidence.
>
> **Q3: Given the definition of ACR, will ACR be provably improved by focusing on easy samples? Is the hypothesis “focusing on easy samples improves ACR” trivially true and does not require empirical evidence in this work? What is the main goal of Algorithm 2?**
>
> The hypothesis is not trivially true. While an improvement on $p_A$ of any input without a decrease on $p_A$ of other inputs leads to increased ACR, it is not guaranteed that improving $p_A$ of easy samples does not reduce $p_A$ of hard samples. In fact, as shown in Sec 4.3, the reverse is usually true. Thus, ACR is not provably increased by focusing on easy samples. Further, the hypothesis will only hold when the benefits of improvement on easy samples exceed the loss on hard samples. This leads to our experimental analysis in Sec. 5 and 6 based on Algorithm 2. The main goal of Algorithm 2 is to both validate this hypothesis and the second hypothesis that the progress in RS training strategies can be replicated by simply focusing on easy samples.
>
> **Q4: Is ACR really misleading, as it is rarely used as a stand-alone metric?**
>
> Please refer to our reply to Q1 of Reviewer $\Rg$.
>
> **Q5: Can the entire paper be summarized “ACR is average of the radii, and averages can be arbitrarily moved by just moving one instance, and hence ACR can be made arbitrarily good by just focusing on easy samples with large robustness radii”?**
>
> This summary is partially true, but misrepresents important scopes established in this work. It is true that ACR is the average of the certified radius, and averages can be arbitrarily moved by an infinite change on only one instance. However, as pointed out in our reply to Q3, this does not directly prove that ACR can be made arbitrarily good by just focusing on easy samples with large radii. In fact, our empirical results in Sec. 6 shows that ACR can at least be amplified to the state-of-the-art (SOTA) by focusing on easy samples, but still not infinitely large. As discussed in Q3, there might exist an equilibrium where the loss on hard samples exactly offsets the improvement on easy samples, leading to maximized ACR. The more accurate summary of this work should be: (i) ACR is problematic because it does not faithfully represents the robustness and it is much more sensitive to the same magnitude of improvements on easy samples than hard samples, (ii) empirical evidence shows that ACR of Gaussian training can be amplified to SOTA by simply focusing on easy samples, and (iii) ACR should be replaced by better alternatives such as certified accuracy at various radii and the ECDF of $p_A$ when evaluating RS training strategies.

---

### Decision · Program_Chairs · 2025-05-01

**Decision:**

Accept (poster)

**Comment:**

The paper investigates the validity of the Average Certified Radius (ACR) as a measure of robustness. The reviewer overall assessed the results positively; however, there are some concerns that need to be addressed, such as supporting some of the claims in the paper by citing relevant references. The paper should include more examples of works in the field and discuss the metrics used by those works to give the reader a better understanding of how commonly the ACR metric is used as the primary evaluation metric in the literature. Including datasets with more natural images is also suggested, and some issues related to easy samples and ACR need to be further discussed.